



**A consistent implementation of the dual node approach for coupling**
**surface-subsurface flow and its comparison to the common node**
**approach**
Rob de Rooij
Water Institute, University of Florida, 570 Weil Hall, PO Box 116601, Gainesville, FL-32611-
6601, USA
r.derooij@ufl.edu
Corresponding author:
Rob de Rooij
Water Institute
University of Florida
570 Weil Hall
PO Box 116601
Gainesville
FL-32611-6601
USA
Telephone: 1-352-392-5893
Fax: 1-352-392-6855



**Key points**
Surface-subsurface flow coupling
**Abstract**
Commonly, the dual node approach for coupling surface-subsurface flow is conceptualized as a
hydraulic separation of the surface and the subsurface by a distinct interface with a given thickness.
Since such an interface is not supported by field observations, it has been argued that the dual node
depends on a non-physical parameter in the form an ill-defined interface thickness. As such, the
alternative common node approach is considered to be a more general and a more elegant approach
since it is based on the physical principle of head continuity along the surface-subsurface interface.
In this study, however, it is argued that if properly implemented, then the dual node approach is
actually the more general, the more elegant as well as the more accurate approach. This insight is
obtained by considering that the topmost subsurface nodal values represent the mean values within
discrete control volumes and by deriving the dual node approach from equations that govern
infiltration and infiltrability. It is shown that the dual node approach should be conceptualized as
a simple one-sided first-order finite-difference to approximate the vertical subsurface hydraulic
gradient at the land surface and that there is no need to assume a hydraulic separation between the
two flow domains by a distinct interface. Whereas a consistent properly implemented dual node
approach is in agreement with the physical principle of head continuity at the land surface, it is
shown that the common node approach is not. Studies that have compared the two coupling
approaches have been based on improperly implemented dual node approaches. As such, this study
presents a re-evaluation of how the common node compares to the dual node approach. Cell-
centered as well as vertex-centered schemes are considered.





## 1 Introduction

There exists a variety of hydrogeological problems, such as the hydrologic response of hill slopes
and river catchments, which requires an integrated analysis of surface and subsurface flows. This
has led to the development of physically-based, distributed parameter models for simulating
coupled surface-subsurface flows. Well-known examples of such models include MODHMS
[*Panday and Huyakorn*, 2004] , InHM [*Ebel et al.*, 2009], HydroGeoSphere [*Therrien et al.*, 2010],
CATHY [*Weill et al.*, 2011] , WASH123D [*Yeh et al.*, 2011], ParFlow [*Kollet and Maxwell*, 2006]
and OpenGeoSys [*Kolditz and Shao*, 2010]. Typically, subsurface flow is governed by the
Richards' equation whereas surface flow is either governed by the kinematic wave or the diffusive
wave equation.
The coupling between subsurface and surface flow may be either based on the common
node approach [*Kollet and Maxwell*, 2006] or on the dual node approach [*Ebel et al.*, 2009; *Panday*
*and Huyakorn*, 2004; *VanderKwaak*, 1999]. In the common node approach coupling is formulated
by a continuity in head between surface and subsurface nodes. The dual node approach is based
on formulating an exchange flux between the surface and subsurface nodes. Typically, the dual
node approach is conceptualized as a hydraulic separation of the surface and the subsurface by a
saturated interface with a given thickness [*Liggett et al.*, 2012]. The thickness of this interface
defines a coupling length between the dual nodes to formulate the discrete exchange flux between
the dual nodes.
It has been argued that the coupling length is a non-physical model parameter, because
there is often no evidence to support the existence of a distinct  interface between the two flow
domains [*Kollet and Maxwell*, 2006]. As such it appears that the common node approach is a more
general coupling approach [*Kollet and Maxwell*, 2006]. Considering that smaller coupling lengths




tend to improve the accuracy of the dual node approach [*Ebel et al.*, 2009; *Liggett et al.*, 2012;
*Liggett et al.*, 2013], it also seems that the common node approach is generally more accurate.
Namely, in the limit as the coupling length goes to zero, the dual node approach mimics the
common node approach [*Ebel et al.*, 2009]. It has been illustrated that both the dual node approach
as well as the common node approach are sensitive to the vertical discretization near the surface
[*Liggett et al.*, 2012; *Sulis et al.*, 2010].

In this study it is illustrated that if the dual node approach is properly implemented as well

as properly conceptualized, then the dual node approach is actually the more general, more elegant
as well as the more accurate approach. This is a significant finding particularly since this
contradicts the findings of other studies in which the common node is commonly regarded as a
more general and more elegant approach [*Dawson*, 2008; *Kollet and Maxwell*, 2006; *Liggett et al.*,
2012; *Liggett et al.*, 2013]. To arrive at a properly implemented or consistent dual node approach
the dual node approach is derived from basic flow equations. Moreover, to develop and understand
the consistent approach, it is crucial to realize that the topmost subsurface nodes should ideally
represent values at the centroids of discrete control volumes. It is shown that the dual node
approach should not be conceptualized as a distinct interface across which an exchange flux is
computed. Instead the dual node approach should be interpreted as a one-sided finite difference
approximation of the vertical hydraulic gradient at the land surface in which the coupling length
is defined by the grid geometry. Moreover, whereas the consistent dual node approach is in
agreement with the principle of head continuity at the surface-subsurface interface, it can be shown
that the common node approach is not.

In this study the coupling approaches are considered for cell-centered as well as vertex-

centered finite difference schemes. Theoretical considerations as well as numerical experiments





indicate that the dual node approach when properly implemented is often more accurate as well as
more computationally efficient than the common node approach, particularly if the vertical
discretization is relatively coarse. This is an important finding because using a relatively coarse
vertical discretization is common practice in regional coupled surface-subsurface models [*Jones*
*et al.*, 2008; *Kollet and Maxwell*, 2008; *Srivastava et al.*, 2014]. The numerical experiments are
carried out with the model code DisCo [*de Rooij et al.*, 2013].
**2   Interpretation of nodal values**
As explained later on, a correct interpretation of nodal values is crucial for understanding the dual
and common node approach for coupling surface-subsurface flow. Moreover, both coupling
approaches depend on the configuration of surface and topmost subsurface nodes near the land
surface. This configuration depends on whether cell-centered or vertex-centered schemes are used.
In this study both type of schemes will be covered, but for simplicity only finite difference schemes
are considered.

In both cell-centered as vertex-centered schemes the flow variables such as the heads and

the saturation are computed on nodes. In vertex-centered schemes these nodes coincide with the
vertices of mesh, whereas in cell-centered schemes the nodes coincide with the cell centers. When
employing a finite difference scheme, nodal values correspond to the mean value within
surrounding discrete control volumes. In cell-centered finite difference schemes these discrete
volumes are defined by the primary grid cells. In vertex-centered finite difference schemes these
discrete volumes are defined by the dual grid cells. Ideally, the mean values in the discrete control
volumes are derived by applying the midpoint rule for numerical integration such that their
approximation is second-order accurate. Therefore, the nodal values should ideally represent





values at the centroid of the surrounding discrete control volume [*Blazek*, 2005; *Moukalled et al.*,
2016]. In that regard, a cell-centered finite difference scheme is thus more accurate than a vertex-
centered finite difference scheme. Namely, in cell-centered finite difference schemes the nodal
values always correspond to the centroids of the cell whereas in vertex-centered finite difference
schemes nodes and centroids (of the dual cells) do not coincide at model boundaries and in model
regions where the primary grid is not uniform.  It is well-known that this mismatch between nodes
and centroids can lead to inaccuracies since the mean values within affected discrete volumes are
not computed by a midpoint rule [*Blazek*, 2005; *Moukalled et al.*, 2016].
Typically, vertex-centered schemes for simulating coupled surface-subsurface flow are
based on mass-lumped finite element schemes [*Liggett et al.*, 2012]and not on finite difference
schemes. However, with respect to coupling surface-subsurface flow there is actually no difference
between a mass-lumped finite element scheme and a vertex-centered finite difference scheme.
Similar as in vertex-centered finite difference schemes, the nodal values in mass-lumped finite
element schemes define the mean values inside dual grid cells [*Zienkiewicz et al.*, 2005].
Moreover, the coupling approaches establish one-to-one relations between surface and topmost
subsurface nodes which do not depend on whether a finite difference or a finite element approach
is being used. Thus, a less complicated vertex-centered finite difference scheme may be used to
provide insights in the coupling approaches as used in mass-lumped finite element schemes.
**3    Common node approach**
The common node approach defines a head continuity between the topmost subsurface nodes and
the surface nodes. This continuity requires that the topmost subsurface nodes and the surface nodes
are co-located at the land surface such that there exists a continuity in the elevation head. This





requirement is automatically full-filled in vertex-centered schemes. Figure 1c illustrates the
configuration of common nodes for vertex-centered schemes. This configuration is similar to the
configuration as used in HydroGeoSphere [*Therrien et al.*, 2010]. However, in cell-centered
schemes such as ParFlow the co-location of nodes is less straightforward. Also, the basic
explanation that the pressure head continuity is assigned at the top cell of the subsurface domain
at the boundary between the two domains [*Kollet and Maxwell*, 2006; *Maxwell et al.*, 2009; *Sulis*
*et al.*, 2010] is ambiguous since the location of the land surface with respect to the top cell is not
specified. Nonetheless, since ParFlow is a cell-centered scheme where the topmost subsurface
node is located at the center of the top cell, it follows that the surface node is located at the center
of the topmost subsurface cells as depicted in Figure 1a such that the land surface is located at the
center of the topmost subsurface cell. This is the correct configuration as applied in ParFlow
[personal communication Maxwell, R. in relation to previous work of the author [*De Rooij et al.*,
2012]]. It can be argued that the additional subsurface volumes that extent above the land surface
do not drastically affect the timing of runoff. Namely, once the topmost subsurface node reaches
fully saturated conditions, the amount of additional water that can be stored in those volumes is
relatively small as long as the specific storage assigned to the topmost cell is relatively small.

Since the location of the land surface in ParFlow is somewhat unclear, some studies have

inferred that ParFlow uses a completely different nodal configuration. For example, it has been
inferred that the topmost subsurface nodes in the ParFlow model are placed on top of the topmost
subsurface cell such that they are co-located with the surface nodes [*Liggett et al.*, 2013]. An and
Yu [*An and Yu*, 2014] infer that the surface and subsurface nodes are not co-located at all and the
surface nodes are located at the top face of the topmost subsurface cells and that the topmost
subsurface nodes are located at the center of the topmost subsurface cells.





Considering that nodal values represent ideally the mean values within discrete control
volumes as described in Section 2, it can be argued that the head continuity as implemented in the
common node approach is not in agreement with the physical principle of head continuity at the
land surface. Namely, the common node approach enforces a continuity between surface heads at
the land surface and the mean subsurface heads within the topmost subsurface discrete control
volumes which have a finite thickness. This is different from enforcing a continuity between
surface heads and subsurface heads within an infinitesimal thin subsurface layer directly below the
land surface. As such inconsistent behavior is expected when using the common node approach.
To effectively remove this inconsistency a very fine vertical discretization is required near the land
surface.
**4    Consistent dual node implementation**
Figure 1b and 1c illustrate the classical arrangement of surface and subsurface nodes in cell-
centered and vertex-centered finite difference schemes, respectively. Commonly, the dual node
approach is expressed in terms of an exchange flux $q_e$ [LT$^{-1}$] computed as [*Liggett et al.*, 2012;
*Panday and Huyakorn*, 2004]:

$$q_e = f_p \frac{K_z}{l} \left( h_s - h_{ss} \right) \tag{1}$$

where $h_s$ and $h_{ss}$ are the hydraulic heads [L] associated with the surface node and the topmost
subsurface node, respectively, $f_p$ [-] the fraction of the interface that is ponded and $l$ the coupling
length [L]. The ponded fraction of the interface is typically defined by a function that varies
smoothly between zero at the land surface elevation and unity at the rill storage height which
defines the minimum water depth for initiating lateral overland flow [*Panday and Huyakorn*,

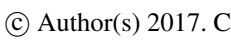


2004]. In equation (1) the term $f_{\mathrm{p}}K_z/l$ is commonly referred to as the first-order exchange
parameter, where first-order means that the exchange flux depends linearly of the hydraulic head
difference.

Typically, equation (1) is not derived as a numerical approximation of basic flow equations

that govern the exchange flux, but is presented a numerical technique to couple two different flow
domains [*Ebel et al.*, 2009; *Liggett et al.*, 2012]. Subsequently, the dual node approach is
conceptualized by interpreting equation (1) as an expression that describes groundwater flow
across a distinct interface separating the two flow domains [*Ebel et al.*, 2009; *Liggett et al.*, 2012;
*Liggett et al.*, 2013]. Evidently, if the coupling length is assumed to be a non-physical parameter,
then it follows that equation (1) cannot be derived from basic flow equations. In the following,
however, it is illustrated that the dual node approach can and should be derived from basic
equations that describe infiltration into a porous medium. This derivation is inspired by but slightly
different from the work of  Morita and Yen [*Morita and Yen*, 2002].

Before deriving the dual node approach from equations that describe infiltration, it is

worthwhile to point out that above formulation of an exchange flux implies that infiltration only
occurs across the ponded fraction of the surface-subsurface interface. This is not correct, because
rainfall typically results in infiltration across non-ponded areas. Although this issue is not a crucial
problem since the ponded fraction will typically increase during rainfall, it is more elegant to
account explicitly for infiltration across non-ponded areas. This is relatively straightforward since
before ponding occurs the infiltration rate equals the rainfall rate if the rainfall rate is smaller than
the infiltrability and is limited to the infiltrability otherwise [*Hillel*, 1982] and such a computation
is also used by others [*Morita and Yen*, 2002].  In the approach presented here the surface cell can




be partially ponded whereas in the work of Morita and Yen [*Morita and Yen*, 2002] a surface cell
is either ponded or non-ponded.

Using Darcy's Law, the infiltration rate at the ponded land surface $q_{s \to ss}$ [LT$^{-1}$] can be

written as a function of the vertical subsurface hydraulic gradient at the land surface:
$$q_{s \to ss} = \left( k_r K_z \frac{\partial h}{\partial z} \right)\bigg|_{z=z_s} = K_z \frac{\partial h}{\partial z}\bigg|_{z=z_s} \qquad (2)$$

where $h$ the hydraulic head [L], $z$ the elevation head [L], $k_r$ the relative hydraulic conductivity [-]
$K_z$ the saturated vertical hydraulic conductivity [LT$^{-1}$] and $z_s$ the elevation head at the land surface.
The relative hydraulic conductivity is unity because equation (2) applies to the ponded land surface
which implies fully saturated conditions at the land surface (i.e. ponding means $p_s > 0$, where $p_s$ is
the pressure head at the surface).  Similarly, the infiltrability [LT$^{-1}$], defined as the infiltration rate
under the condition of atmospheric pressure [*Hillel*, 1982], can be written as:
$$I = \left( k_r K_z \frac{\partial h}{\partial z} \right)\bigg|_{z=z_s, p_s=0} = K_z \frac{\partial h}{\partial z}\bigg|_{z=z_s} \qquad (3)$$

The relative hydraulic conductivity is again unity because the saturation equals unity under
atmospheric conditions ($p_s = 0$). The infiltration rate at non-ponded land surface $q_{atm \to ss}$ [LT$^{-1}$] can
be expressed as:
$$q_{atm \to ss} = \min \left( \max \left( I, 0 \right), q_R \right) \qquad (4)$$

where $q_R$ is the effective rainfall rate (i.e. the infiltration rate is limited by either the infiltrability
or the available effective rainfall rate). The total exchange flux across the surface-subsurface
interface can now be written as:
$$q_e = f_p q_{s \to ss} + \left( 1 - f_p \right) q_{atm \to ss} \qquad (5)$$




To approximate the vertical subsurface hydraulic gradient in equations (2) and (3), it is
crucial to recognize that according to the principle of head continuity at the land surface, the
surface hydraulic head at a surface node must also represent the subsurface head at the land surface
at that location. Thus, the surface hydraulic head can be used as a Dirichlet boundary condition for
the subsurface flow domain. Moreover, it is also crucial to recognize that since the subsurface
hydraulic heads at the topmost subsurface nodes are ideally associated with the centroids of the
topmost subsurface discrete control volumes, these head values do not represent values at the land
surface but at some depth below the land surface. Because the subsurface hydraulic heads at the
dual nodes can be and should be associated with a different elevation, the vertical subsurface head
gradient between the dual nodes can be approximated by a standard finite difference
approximation. If this approximation is being used to approximate the gradient at the land surface
in equations (2) and (3), then this approximation is by definition a one-sided first-order finite
difference. Defining the coupling length by $l = \Delta z$ where $\Delta z$ is the difference in the mean elevation
head associated with the dual nodes, the infiltration rate and infiltrability can thus be computed
with the following one-sided finite difference approximation:
$$K_z \left. \frac{\partial h}{\partial z} \right|_{z=z_s} \approx \frac{K_z}{l}\left(h_s - h_{ss}\right) \tag{6}$$

The above definition of the coupling length $l = \Delta z$ ensures a proper approximation of the vertical
gradient in elevation head at the land surface:
$$\left. \frac{\partial z}{\partial z} \right|_{z=z_s} = \frac{\Delta z}{l} = 1 \tag{7}$$

Since nodal values in cell-centered scheme are located at the centroids of the cells, the coupling
length is simply given by $l = z_s - z_{ss}$. This value has been proposed by others [*Panday and*




*Huyakorn*, 2004]. However, in vertex-centered schemes the commonly used nodal configuration
near the surface is such that $z_s = z_{ss}$. If these elevation heads are used as the elevation heads at the
dual nodes then $\Delta z = z_s - z_{ss} = 0$. Since the coupling length must be greater than zero, the coupling
length cannot be defined as $l = \Delta z$. Indeed, the coupling length in vertex-centered schemes is
typically not related to grid structure [*Liggett et al.*, 2013]. However, if $\Delta z = 0$ and the coupling
length is some lumped-parameter greater than zero, then the dual node approach is inconsistent.
Namely, if $\Delta z = 0$ then the gradient in elevation head between the dual nodes equals zero. This
may seem correct as the nodes are co-located. However, if $z_{ss} = z_s$, then the physical principle of
head continuity implies that $p_{ss} = p_s$ must also hold. Moreover, even though the topmost
subsurface node is located at the land surface in a vertex-centered scheme, the elevation head at
this node should ideally correspond to the mean elevation head within the topmost subsurface
discrete control volume such that $z_{ss} < z_s$. This suggests that the topmost subsurface node should
be moved to the centroid of the topmost subsurface discrete volume. Although this is a possible
solution, the drawback of this solution is that the subsurface model ceases to be a purely vertex-
centered scheme. Moreover, such an operation cannot be performed in finite element schemes
since changing the nodal positions would change the elements. Therefore, an alternative solution
is proposed. To enforce $l = z_s - z_{ss}$ without affecting the relative positions of nodes in the
subsurface grid, the elevation of the surface nodes are changed according to $z_s = z_{ss} + l$ where $l$ is
equals half the thickness of the topmost subsurface dual cell. This change does also not affect the
relative position of the nodes in the surface grid. The resulting nodal configuration is illustrated in
Figure 1d. In essence, the motivation behind this solution is that a more accurate approximation
the hydraulic gradient is more important than the actual elevation of the land surface. Indeed it can





be argued that the change in land elevation will not drastically affect the timing of runoff. Namely,
once the topmost subsurface node reaches fully saturated conditions, the amount of additional
water needed to reach the elevated land surface is minor as long as the specific storage assigned to
the topmost dual cell is relatively small.

It is crucial to observe that the proposed dual node implementation is not based on

assuming a distinct interface with a certain thickness between the subsurface and the surface.
Instead, the coupling length is to be interpreted as a distance between dual nodes that accounts for
the fact that the topmost subsurface nodal value ideally corresponds to a value below the land
surface. This distance is related to the vertical discretization near the land surface and as such does
not represent a non-physical parameter associated with a distinct interface separating the two
domains.

The common conceptualization of the dual node approach as a hydraulic separation by a

interface with a given thickness [*Kollet and Maxwell*, 2006; *Liggett et al.*, 2012; *Liggett et al.*,
2013], may arise if dual node approach is interpreted as a second-order central finite difference
approximation evaluated at the centre of a saturated layer with a thickness equal to the coupling
length. If in addition the topmost subsurface head values are taken as values at the land surface,
then it follows that the dual node approach introduces a distinct interface between the two flow
domains. However, as explained the topmost subsurface head values should not be taken as values
at the land surface but as values at some distance from the land surface, such that the interface
defined by the coupling length occupies the upper half of the topmost subsurface discrete control
volumes.

It is also worthwhile to explain in further detail that the dual node approach does not

account for the relative hydraulic conductivity near the land surface. This does not imply that the





subsurface near the land surface is saturated. Namely, saturation in the topmost subsurface discrete
volume is computed with the pressure head at the topmost subsurface node which may well be
below zero. It may appear that the vertical hydraulic conductivity between the dual nodes should
be computed by weighting the vertical hydraulic conductivities at the dual nodes, which would
result in a dependency on the relative hydraulic conductivity as long as the topmost subsurface
node is not fully saturated. However, no weighting is needed if the dual node approach is
understood as a one-sided finite difference evaluated at the land surface. Namely, the vertical
hydraulic conductivity at the land surface is readily available. This is a difference with respect to
the approach of Morita and Yen [*Morita and Yen*, 2002] who do use a weighting scheme.
Moreover, models typically apply upstream weighting to approximate the relative hydraulic
conductivities between nodes to avoid numerical instabilities [*Forsyth and Kropinski*, 1997]. Thus
even if weighting is applied, then the dependency of the computations between the dual nodes on
the relative hydraulic conductivity will automatically disappear as the upstream node is always
saturated.

To illustrate that the presented dual node approach exhibits consistent behaviour, the

necessary conditions for ponding due to excess infiltration and exfiltration are considered. In
general ponding starts when $q_R > I$ [*Hillel*, 1982]. Setting $q_R = I$, $p_s = 0$ and using $h = p + z$, it
follows from equation (6) and (7) that at the moment of ponding:

$$p_{ss} = l\left(1 - \frac{q_R}{K_z}\right) \qquad (8)$$

Ponding due to excess infiltration occurs if $q_R / K_z > 1$ and implies that saturation in the subsurface
starts from the top down [*Hillel*, 1982]. Using $q_R / K_z > 1$ it follows from equation (8) that ponding
due to excess infiltration occurs while $p_{ss} < 0$. This is reasonable since this value represents the



pressure head at a certain depth below the land surface. Namely, if saturation occurs from the top-
down then the saturation at a certain depth occurs later than saturation at the land surface. It is
noted that if the ratio $q_R / K_z$ is greater than but close to unity or if the coupling length is very small,
then this condition becomes $p_{ss} \approx 0$. Once ponding starts the total flux rate between the dual nodes
equals $K_z \left( (p_s - p_{ss})/l + 1 \right)$. Top-down saturation requires that this flux exceeds the vertical
hydraulic conductivity. Reaching saturation at the topmost node ($p_{ss} = 0$) thus requires $p_s \geq 0$.
Thus, top-down saturation will occur after ponding is initiated. Ponding due to excess saturation
occurs if $q_R / K_z < 1$ and implies that saturation in the subsurface starts from the bottom up [*Hillel*,
1982]. Using $p_s = 0$, it follows from equation (8) that ponding due to excess saturation occurs while
$0 < p_{ss} < l$. Thus ponding starts after reaching fully saturated conditions at the topmost subsurface
node, which is again reasonable. Namely, the topmost subsurface node represents a value at a
certain depth below the surface and thus bottom-up saturation implies that this node reaches
saturation earlier than the surface. It is noted that if the ratio $q_R / K_z$ is smaller than but close to
unity or if the coupling length is very small, then ponding occurs when $p_{ss} \approx 0$.
**5     Comparison to other dual node implementations**
To illustrate that it is crucial to account for the meaning of the values at the topmost subsurface
nodes, it is instructive to consider what happens if these values are not taken as the mean values
within discrete control volumes. As a first example, consider vertex-centered schemes where the
dual nodes are defined such that $z_{ss} = z_s$ as illustrated in Figure 2c. As discussed in Section 4 this
is inconsistent because it defines a zero gradient in elevation head between the dual nodes.
Nonetheless such schemes have been used in several models [*Ebel et al.*, 2009; *Liggett et al.*,



2012]. Since the vertical gradient in elevation head between the dual nodes is zero the total flux
rate after ponding now equals $K_z\left(p_s - p_{ss}\right)/l$. Top-down saturation requires that this flux exceeds
the vertical hydraulic conductivity. Thus, reaching saturation at the topmost subsurface node (
$p_{ss} = 0$) requires $p_s > l$. Therefore, top-down saturation will not occur if runoff occurs and if the
surface water depths remains smaller than the chosen coupling length. Indeed, it has been pointed
out in other studies that the coupling length should be smaller than the rill storage height [*Delfs et*
*al.*, 2009; *Liggett et al.*, 2012]. The zero vertical gradient in elevation head between the dual nodal
also means that the required condition for ponding now becomes $p_{ss} = -lq_R / K_z$. This implies that
ponding due to excess saturation occurs while the topmost subsurface node is not yet saturated.

A second example is the dual node approach for cell-centered schemes as implemented in

MODHMS which uses an adapted pressure-saturation relationship for the topmost subsurface
nodes such that the topmost subsurface node only becomes fully saturated if hydraulic head at the
node rises above the land surface [*Liggett et al.*, 2013]. Since the topmost subsurface heads are
associated with the cell centroid, this dual node scheme defines a unit gradient in elevation head
at the land surface. However, the saturation value at the topmost node is associated with a location
at the land surface and not with the centroid of a discrete control volume. This has undesirable
consequences. Namely, saturating the topmost subsurface node ($p_{ss} = l$) due to excess infiltration
requires that $p_s > l$. Indeed, when simulating excess infiltration with MODHMS, a very small
coupling length is needed to simulate top-down saturation due to excess infiltration. [*Gaukroger*
*and Werner*, 2011; *Liggett et al.*, 2013]. It can also be shown that ponding due to excess saturation
occurs while $0 < p_{ss} < l$. But, because of the adapted pressure-saturation relationship this means
that ponding starts while the topmost subsurface node is not yet saturated. Comparing these results



with the results for the consistent dual node implementation, it is clear that the adapted pressure-
saturation relationship has undesirable consequences.

The above inconsistent implementations of the dual node approach have been used in several

studies to compare the dual node approach with the common node approach [*Liggett et al.*, 2012;
*Liggett et al.*, 2013]. Such studies indicate that the dual node approach is typically only competitive
with the common node approach in terms of accuracy once the coupling lengths are very small.
The requirement for very small coupling lengths, however, are a direct consequence of using
inconsistent dual node approaches. Namely, by choosing very small coupling lengths these
inconsistencies are to some extent minimized. At best this minimization results in schemes that
mimic the common node approach.  However, as discussed, the common node approach is also
inconsistent since it is not in agreement with the physical principle of a head continuity at the
surface-subsurface interface. Since current views on how the coupling approaches compare are
based on inconsistent dual node approaches, it is imperative to re-evaluate how the dual and
common node approaches compare if the dual node approach is properly implemented.

Considering how the dual and the common node approach compare it is also crucial that the

dual node approach is not to be conceptualized as a hydraulic separation between the flow domains
in the form of a saturated interface. Namely, this conceptualization is often deemed a serious
drawback of the dual node approach, since there is no evidence of such a distinct interface.
Moreover, misconceptions about the coupling approaches can result in confusion. For example, in
their paper An and Yu [*An and Yu*, 2014] reject the idea of using the dual node based on its classical
conceptualization as a saturated interface and argue that their model is based on the approach
proposed by Kollet and Maxwell [*Kollet and Maxwell*, 2006]. However, in their finite volume
model the surface and subsurface nodes are not co-located. As such their coupling approach is,





contrary to the claim of the authors, a dual node approach. This misunderstanding is probably also
related to aforementioned difficulties in inferring the nodal configuration as used in ParFlow.
Nonetheless, their approach is actually a properly implemented dual node approach practically
similar to the one proposed in this paper. Interestingly, the model of An and Yu [*An and Yu*, 2014]
is less sensitive to the vertical discretization near the land surface in comparison to ParFlow
However, since An and Yu were convinced that they followed the same coupling approach as
ParFlow they hypothesized that the difference in performance was probably related to using
irregular grids instead of orthogonal grids as in ParFlow [*An and Yu*, 2014]. However, if this
difference is instead due to using a different coupling approach, then this would be an indication
that a dual node approach is less sensitive to the vertical discretization near the land surface. This
reinforces the idea that it is desirable to reconsider the comparison between the two coupling
approaches.
**6    Numerical experiments**
To compare the coupling schemes in terms of accuracy and computational efficiency numerical
experiments are presented. These experiments are carried out with the model code DisCo which
can simulate coupled surface-subsurface flow using a fully implicit or monolithic scheme [*de Rooij*
*et al.*, 2013]. This means that the linearized surface and subsurface flow equations are combined
into a single matrix system. An adaptive error-controlled predictor-corrector one-step Newton
scheme [*Diersch and Perrochet*, 1999] is used in which a single user-specified parameter controls
the convergence as well the time stepping regime. It is assumed that by using the same error norms
and the same model parameters that control the time-stepping, the simulations results as obtained
by different coupling approaches can be compared fairly in terms of accuracy and efficiency. For





brevity further details about the model are not discussed here and can be found elsewhere [*de Rooij*
*et al.*, 2013].

Table 1 lists the abbreviations used in the figures to distinguish between the coupling

approaches, to distinguish between cell-centered and vertex-centered schemes and to distinguish
between models based on a uniform primary grid and grids that use a very thin primary top cell.
The thickness of this top cell equals the thickness of the primary cells in the finest uniform grids.
In models containing this thin layer of cells the vertical discretization below the thin layer is based
on the coarsest uniform grids. Further details about the discretizations are given in the figures.

The presented experiments focus mainly on the comparison between the consistent dual

node approach and the common node approach. Inconsistent dual node implementations based on
a zero hydraulic head gradient between the dual nodes are only considered for relatively coarse
vertical discretizations to illustrate their short-comings vis-à-vis the consistent dual node approach.
It is noted, that although these schemes are commonly used in vertex-centered schemes, for the
purpose of this study they have also been implemented in the cell-centered schemes by using the
nodal configuration depicted in Figure 1a. The scheme with an adapted pressure-saturation
relationship is not considered.

## 6.1    Soil column problems

These simulation scenarios consider infiltration into a vertical soil column and are inspired by
scenarios as studied by Liggett et al. [*Liggett et al.*, 2012; *Liggett et al.*, 2013]. In the simulation
scenarios rainfall is applied to a soil column with a height of 5 m. Initial conditions are defined by
$h = 0$ m. The saturated conductivity is 1.0608 md$^{-1}$. The porosity is 0.41 and the specific storage
is 10$^{-4}$ m$^{-1}$. The van Genuchten parameters are given by $s_r = 0.387$, $s_s = 1.0$, $\alpha = 7.5$ m$^{-1}$ and $n =$

1.89.



For the first two scenarios a constant head boundary of $h = 0$ m is applied at the bottom of
the column and the flux rate applied to the top of the soil column exceeds the saturated conductivity
of the soil column, resulting in runoff due to excess infiltration. In the first scenario the applied
flux rate is 1.1 md$^{-1}$. Figure 2 and 3 illustrates the simulated runoff and the number of Newton
steps for this scenario, respectively. Figure 4 illustrates the simulated runoff for the second
scenario in which the flux rate is 10.608 md$^{-1}$. It is noted that figure 4 does not display the results
at later times when a steady-state is reached. However, to show the differences in results around
the timing of ponding only a limited time period is displayed. Figure 5 illustrates the number of
Newton steps for the second scenario. For the second scenario, Figure 6 compares the evolution in
water depth between the common node approach and the dual node approach when using a
relatively coarse vertical discretization and a cell-centered scheme.
To compare the different coupling approaches when simulating excess saturation, a third
scenario is considered. The model setup is exactly the same as before, except that the effective
rainfall rate is set to 0.5 md$^{-1}$ and that the bottom boundary is changed into a no-flow boundary.
The simulated runoff is depicted in Figure 7. Figure 8 shows the total number of Newton steps
during the model runs. Figure 9 compares the evolution in water depth between the common node
approach and the dual node approach when using a relatively coarse vertical discretization and a
cell-centered scheme.
**6.2    Hillslope problems**
In the following the first two simulation scenarios consider hillslope problems as designed by Sulis
et al. [*Sulis et al.*, 2010]. For the purpose of this study, a third scenario is considered in which the
initial and boundary conditions are different to create a flooding wave across an unsaturated
hillslope. The problems consist of a land surface with a slope of 0.05 which is underlain by a



porous medium. The domain is 400 m long and 80 m wide. The subsurface is 5 m thick. In the

direction of the length and in the direction of the width the discretization is 80 m. Different vertical

discretizations are considered. The van Genuchten parameters are given by $s_r = 0.2$, $s_s = 1.0$, $\alpha = 1$

$m^{-1}$ and $n = 2$. The porosity is 0.4 and the specific storage is $10^{-4}$ $m^{-1}$. The manning's roughness

coefficients are given by 3.3 x $10^{-4}$ $m^{-1/3}$min. The surface flow domain has a zero-gradient outflow

condition. For the first two simulation scenarios the domain is recharged with an effective rainfall

rate of 3.3 x $10^{-4}$ m/min for a duration of 200 minutes and the initial water table depth is at a depth

of 1.0 m below the land surface.

The first scenario considers excess infiltration and the saturated hydraulic conductivity

equals 6.94 x $10^{-6}$ m/min. Figure 10 and 11 show the simulated runoff and the number of Newton

steps, respectively. For the second scenario which considers excess saturation, the saturated

conductivity equals 6.94 x $10^{-4}$ m/min. Figure 12 and 13 illustrates the simulated runoff and the

number of Newton steps, respectively. In the third scenario a surface water flood wave crossing

the hillslope in the downhill direction is simulated by applying a Neumann boundary condition of

1.0 $m^3$/s to the surface nodes with the highest elevation. The initial water table is located at a depth

of 1.5 m. The vertical saturated hydraulic conductivity equals 6.94 x $10^{-6}$ m/min. Figure 14

illustrates the differences in simulated runoff and Figure 15 illustrates the number of Newton steps

of the model runs. Figure 16 compares the evolution in water depth on the surface nodes as well

as the time step sizes between the common node approach and the dual node approach when using

a relatively coarse vertical discretization and a cell-centered scheme.





## 7 Discussion

### 7.1 Accuracy

Considering the simulation of vertical flow through the unsaturated zone, a relatively fine vertical discretisation is needed to simulate sharp saturation fronts with the Richards' equation [*Pan and Wierenga*, 1995; *Ross*, 1990]. A relatively fine vertical discretisation also implies that the common node approach will be in close agreement with the physical principle of head continuity along the surface-subsurface interface. Finally, if the vertical discretisation is relatively small then the coupling length for the consistent dual node approach is also small and this implies that the dual node approach mimics the common node approach. Therefore, it is expected that the coupling approaches will give similar and accurate results if the vertical discretization is sufficiently fine. Indeed, the simulations results indicate that a relatively fine and uniform vertical discretization yields similar results for the common node approach as well as for the consistent dual node approach (Figure 2a, 4a, 5a, 7a, 10a, 12a and 14a). The simulation results based on the finest vertical discretization may thus be taken as reference solutions that enables a comparison of the coupling approaches when a coarser vertical discretization is used. This is an important issue, because using a relatively coarse vertical discretization is common practice in regional coupled surface-subsurface models [*Jones et al.*, 2008; *Kollet and Maxwell*, 2008; *Srivastava et al.*, 2014].

### 7.1.1 Excess saturation

The simulation results of runoff due to excess saturation as obtained by the common node approach and the consistent dual node approach illustrate that simulating excess saturation runoff is not significantly affected by the vertical discretization (Figure 7 and 12). This is because the time needed to reach fully saturated conditions in the subsurface is a simple function of the flow boundary conditions and the initial water content. It is thus expected that the vertical discretization



does not significantly affect the simulation of excess saturation. Although the vertical
discretization may affect the computed initial water content, this effect is usually negligible. It has
been found in other studies that the vertical discretization has little effect on simulated runoff due
to excess saturation [*Sulis et al.*, 2010].

As described in Section 4, when using the consistent dual node approach, ponding due to

excess saturation occurs when $0 < p_{ss} < l$. Thus at the moment of ponding the hydraulic head at
the topmost subsurface node is generally below the land surface. When using the common node
approach, the hydraulic head at the topmost subsurface node is at the land surface at the moment
of ponding. However, if the specific storage is relatively small, then the timing of runoff will be
similar for both coupling approaches. Both approaches are thus expected to yield similar and
reasonably accurate results even when the vertical discretization is relatively coarse. Indeed, the
simulation results indicate that there is little difference between the common node approach and
the consistent dual node approach (Figure 7 and 12).

As indicated in figure 7d, when using an inconsistent dual node approach, the timing of

runoff may be underestimated unless a very small coupling length is being used. As discussed in
section 5 this is expected.
**7.1.2   Excess infiltration**
When simulating excess infiltration the common node approach requires fully saturated conditions
at the topmost subsurface node for ponding to occur. This is a direct consequence of the head
continuity between the surface nodes and the topmost subsurface nodes. However, top-down
saturation associated with excess infiltration implies that reaching fully saturated conditions in the
topmost subsurface discrete volumes should requires more time than reaching fully saturated
conditions in the very near surface, especially if the vertical discretization is relatively coarse. It is



thus expected that the common node approach delays runoff and this delay increases for a coarser
vertical discretization. In addition, if the saturation fronts are less sharp due to a relatively coarse
vertical discretization, it takes more time to reach saturated conditions at the common node. This
will further delay runoff. Indeed, the simulation results indicate clearly that runoff is delayed when
using the common node approach, particularly if the vertical discretization is relatively coarse
(Figure 2, 4, 10 and 14). It has also been found in other studies that the common node approach
delays runoff due to excess infiltration if the vertical discretization is relatively coarse [*Sulis et al.*,
2010]. The overestimation of the infiltration associated with the delay in runoff may result in runoff
due to excess saturation even if the applied flux rate should result in runoff due to excess
infiltration. This is illustrated in Figure 10c for the model run based on a cell-centered scheme and
the common node approach. This Figure illustrates that overestimating the infiltration can yield a
distinctive higher peak in runoff. Comparing this peak with the runoff responses in Figure 12, it is
clear that this model run simulates runoff due to excess saturation

In comparison, the consistent dual node displays more desirable behaviour. Namely, as

explained in Section 4, ponding due to excess infiltration occurs before reaching fully saturated
conditions at the topmost subsurface node which is arguably more correct if saturation occurs from
the top-down, particularly if the vertical discretization is relatively coarse. When using the
consistent dual node approach, the moment of ponding depends on the computation of the
infiltrability. A relatively coarse vertical discretization may result in an underestimation of the
vertical pressure gradient at the land surface. This is because in a soil close to hydrostatic
conditions the pressure heads increase with depth. Therefore the infiltrability during the early
stages of infiltration may be underestimated. If the applied flux rate is sufficiently large, then this
underestimation will result in an underestimation of the timing of runoff. It may be observed from



equation (8) that if the ratio $q_R/K_z$ or the coupling length is sufficiently large, then ponding is
initiated immediately. Figure 10c and 14c illustrate that the timing of runoff can indeed be
underestimated due to a relatively coarse vertical discretization when using the consistent dual
node approach. However, during the later stages of infiltration the pressure head at the topmost
subsurface node will be underestimated due to the combined effect of an underestimated
infiltration rate and the overly diffused saturation fronts. This results in an overestimation of the
infiltration rate in the later stages. Thus at some time after ponding has started, it is expected that
the amount of runoff is underestimated. Contrary to the common node approach, however, there
will be a time at which runoff is simulated correctly (Figure 10c and 14c).

If the applied flux rate is not sufficiently large, then the underestimated infiltrability in the

early stages of infiltration will not be exceeded. In that case, the overly diffused saturation fronts
resulting from a relatively coarse vertical discretization will eventually lead to an underestimation
of pressure head at the topmost subsurface node and as such the infiltrability may be overestimated
at later times. Consequently, when using the consistent dual node approach the timing of runoff
due to excess infiltration may also be underestimated. As discussed in section 4 if the ratio $q_R/K_z$
goes to unity, then the consistent dual node approach behaves practically similar to the common
node approach. Indeed, Figure 2b which depicts a simulation with a relatively small ratio $q_R/K_z$
clearly illustrates that the timing of runoff may be underestimated when using the consistent dual
node approach. However, the delay in runoff as simulated by the consistent dual node approach
will only equal the delay in runoff as simulated by the common node approach in the limit when
$q_R/K_z$ goes to unity. In general, if the consistent dual node approach delays runoff, this delay will
be smaller than the delay in runoff as simulated by the common node approach (Figure 2b).
Overall, regardless if the consistent dual node approach underestimates of overestimates the timing



of runoff, the simulation results indicate that the consistent dual node approach is generally less
inaccurate than the common node approach for simulating excess infiltration when using a
relatively coarse uniform vertical discretization.
As illustrated in Figure 2b, 4b, 10b and 14b, if the coupling approach and the vertical
discretization are identical and if the thin layer is absent, then the vertex-centered schemes are
more accurate with respect to the cell-centered schemes. This difference in accuracy results solely
from the fact the primary mesh is the same for both schemes. As such the vertical extent of the
topmost subsurface volumes is twice as small when using the vertex-centered scheme. This
difference in vertical grid resolution near the land surface explains the differences in accuracy
between the schemes.
When using a thin layer at the top of the model the common node approach and consistent
dual node approach provide similar simulation results as shown in Figure 2c, 4c, 10d and 14d. This
is expected, because the thin layer implies a small coupling length and as such the consistent dual
node approach mimics the common node approach. In essence, in schemes using the consistent
dual node approach the thin layer establishes a near head continuity between the dual nodes. If the
simulation results are compared to the models based on the coarsest uniform discretization (Figure
2b, 4b, 10c and 14c), it is observed that adding a thin layer has only a positive effect on the cell-
centered schemes based on the common node approach. This positive effect is explained by the
fact that due to the thin layer the common node approach is in almost full agreement with the
principle of head continuity at the land surface. Vis-à-vis the corresponding model without a thin
layer, the thin layer has a negligible effect on the cell-centered scheme based on the consistent dual
node approach. This is because the thin layer establishes a head continuity between the dual nodes
and the topmost subsurface node and the adjacent subsurface node below act like the dual nodes



in the model without the thin layer. The thin layer has also a negligible effect on the vertex-centered
scheme based on the common approach. In this case the thin layer establishes a near head
continuity between the topmost subsurface node and the adjacent node below and ponding due to
excess infiltration will require almost fully saturated conditions in the two topmost subsurface
volumes. The sum of these two volumes is equal to the topmost volume in the model without the
thin layer and therefore the effect of the thin layer is minimal.  In a vertex-centered scheme based
on the consistent dual node approach, the thin layer has a clear negative effect. In essence the head
continuity between the dual nodes removes the benefits of using the consistent dual node approach
and contrary to the cell-centered scheme based on the consistent dual node approach the topmost
subsurface node and the adjacent subsurface node below do not act like the dual nodes in the model
without the thin layer. This is because the thin layer creates a non-uniform primary mesh in which
the subsurface node directly below the topmost subsurface node is not located at the centroid of
its associated dual cell.
As indicated in figure 2d and 4d, when using an inconsistent dual node approach, the runoff
is overestimated unless a very small coupling length is being used. As discussed in section 5, this
is expected.

### 7.2 Computational efficiency

During the early stages of ponding the rates at which the water depths are changing can be
relatively fast as the applied flux rates on the land surface are possibly quite large. Typically, a
numerical model with adaptive time-stepping will decrease the time step size at the moment of
ponding to handle the non-linear flow terms and the high rates of change in water depth. Since a
higher infiltration rate at the moment of ponding results in lower initial rates of change in water





depth, it is expected that the most efficient coupling approach is characterized by a higher
infiltration rate at the moment of ponding.

The computational efficiency of the schemes is measured in terms of the number of Newton

steps. The number of Newton steps equals the number of times that the linearized system of
equations is solved and this number depends on the time step sizes as well as the number of failed
Newton steps.

### 7.2.1   Excess saturation


When simulating excess saturation the subsurface is fully pressurized at the moment of ponding
and can only accommodate additional water volumes by means of the specific storage.  As such
the column will be close to hydrostatic conditions at the moment of ponding. When using the
common node approach this implies that the hydraulic gradient between the common node and the
adjacent subsurface node below is very close to zero. When using the consistent dual node
approach ponding due to excess saturation occurs when $0 < p_{ss} < l$. Thus, at the moment of
ponding the hydraulic head at the topmost subsurface node is generally still below the land surface.
This means that the infiltration rate at the moment of ponding as computed by the consistent dual
node approach is higher in comparison to the rate as computed by the common node approach. It
is thus expected that the consistent dual node approach is more efficient when simulating excess
saturation.  Indeed, Figure 8 and 13 illustrate that, when simulating excess saturation, the
consistent dual node approach is more efficient then the common node approach. Figure 10
illustrates the pressure heads on the nodes near the land surface as simulated by the models based
on the cell-centered scheme and the coarsest vertical discretization. As illustrated, the pressure
head gradient governing the infiltration rate at the moment of ponding is larger when using the
consistent dual node approach and consequently the rate of change in water depth is smaller.





### 7.2.2 Excess infiltration

As discussed in section 7.1.2, in comparison to the consistent dual node approach, the common node approach yields a later time of ponding due to excess infiltration. Since saturation fronts in a homogeneous medium become more diffused with time, it follows that the common node approach yields a smaller infiltration rate at the moment of ponding. Namely, if the saturation fronts are more diffused, then the pressure head gradient governing the infiltration rate is less sharp. Therefore, it is expected that the common node approach is computationally less efficient than the consistent dual node approach, particularly if ponding is significantly delayed. Figure 5 illustrates clearly, that the consistent dual node approach can be more computationally efficient. For the simulation scenario depicted in Figure 5, the consistent dual node approach is also more accurate. Figure 4b illustrates that, compared to the consistent dual node approach, the common node approach can result in a relatively high rate of change in runoff at the moment of ponding. This is indicative of a relatively high initial rate of change in water depth at the moment of ponding. Figure 7 illustrates the pressure heads at the nodes near the land surface as simulated by the cell-centered schemes based on the coarsest vertical discretization. It can be observed that the pressure head gradient at the moment of ponding is larger when using the consistent dual node approach. This implies a higher infiltration rate and a lower rate of change in water depth. Figure 11 also illustrates that the consistent dual node approach is more efficient when handling the activation of ponding. However, considering the entire simulation period, the dual node approach is not always more efficient. As illustrated by Figure 11b and 11c, when the discretization is relatively coarse the common node approach is sometimes more efficient during the later stages of the simulation. However, in these cases the common node approach is only more efficient, because its inaccuracy



leads to an easier flow problem to be solved. Namely, the underestimation of runoff results in more
diffused saturation fronts in the subsurface.
Figure 2, shows that if the ratio $q_R/K_z$ is relatively small, then the differences in
computational efficiency are relatively small. As discussed in section 4 this is because the
consistent dual node approach behaves very similar to the common node approach if the ratio $q_R/K_z$
is relatively small.
Another factor that affects the efficiency of the common node approach is that the delay in
ponding can act as an artificial barrier for a surface water wave advancing across an initially
unsaturated subsurface domain. The effect of this artificial barrier is that the front of the surface
water wave is steepened. This steepening of the surface wave front results in higher rates at which
the water depth is changing and is undesirable because it decreases the computational efficiency.
This is clearly illustrated in Figure 15. Figure 16 illustrates the evolution of water depth at the land
surface for the cell-centered schemes using the coarsest vertical discretization. As shown, the
common node approach delays and steepens the surface water front. This results in relatively high
rates of change in water depth at the moment of ponding. Consequently, the common node
approach is less efficient than the dual node approach. It is noted that for this scenario the
consistent dual node approach is more efficient as well as more accurate.
**8    Conclusions**
In this study it is shown that contrary to the common held view, the dual node approach if properly
implemented is actually the more general, the more elegant as well as the more accurate coupling
approach in comparison to the common node approach. This consistent dual node approach is
implemented in cell-centered as well as vertex-centered finite difference schemes.



The consistent dual node approach is derived from basic equations that govern infiltration
and infiltrability at the land surface using a one-sided finite differences approximation of the
vertical hydraulic gradient at the land surface. In both cell-centered as vertex-centered schemes
the coupling length is related to the grid geometry. As discussed, the dual node approach should
not be conceptualized as a distinct interface between the surface and the subsurface. Moreover,
this approach is in agreement with principle of head continuity along the land surface whereas the
common node approach is not, unless the vertical discretization is sufficiently fine.
Numerical experiment indicate that if the vertical discretization is relatively coarse, then
the consistent dual node approach is often less inaccurate as well as more computationally efficient
in comparison to the common node approach for simulating excess infiltration. For simulating
excess saturation both coupling approaches are more or less equally accurate, but the consistent
dual node approach was found to be more computationally efficient. Therefore, overall it can be
argued that the consistent dual node approach is to be preferred to the common node approach
unless the vertical discretization is sufficiently fine such that both approaches yield similar results.

**Acknowledgements**
This research was funded by the Carl. S. Swisher Foundation.

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

| abbreviation | meaning |
| --- | --- |
| cc | cell-centered |
| vc | vertex-centered |
| dn | dual node |
| cn | common node |
| TL | tiny layer |


Table 1: Abbreviations as used in the figures.




a)                b)








c)                d)














Figure 1: a) Common nodes and co-located dual nodes in cell-centered schemes. b) Common nodes and
co-located dual nodes in vertex-centered schemes. c) Dual nodes in cell-centered-centered schemes (not
col-located). d) Dual nodes in vertex-centered schemes (not co-located). The white squares and white
circles represent surface and subsurface nodes, respectively. The solid and dashed lines represent the
primary mesh and the dual mesh, respectively. The grey-shaded area is a topmost discrete volume as
associated with a topmost subsurface node. The black dot represents the centroid of this volume. The
coupling length $l$ as depicted in this figure applies to the consistent dual node approach.












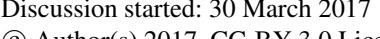



Figure 2: Simulated runoff for excess infiltration in a vertical soil column using different vertical discretizations ($q_R = 1.1$ md$^{-1}$).



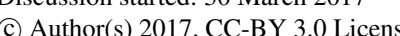



Figure 3: Number of Newton steps for excess infiltration in a vertical soil column using different vertical

discretizations ($q_R$ = 1.1 md$^{-1}$).






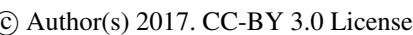

Figure 4: Simulated runoff for excess infiltration in a vertical soil column using different vertical discretizations ($q_R$ = 10.608 md$^{-1}$).



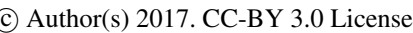



Figure 5: Number of Newton steps for excess infiltration in a vertical soil column using different vertical discretizations ($q_R$ = 10.608 md$^{-1}$).

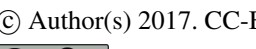



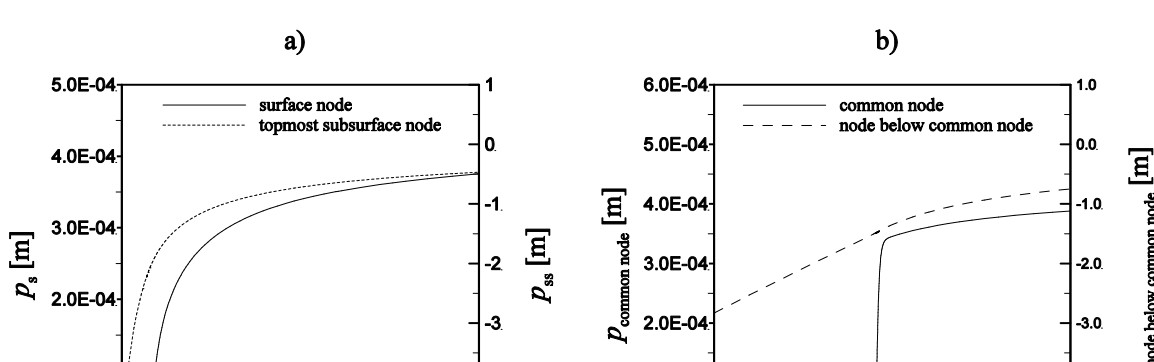


Figure 6: Changes in pressure heads near the surface-subsurface interface for excess infiltration in a vertical
soil column ($q_R$ = 10.608 md$^{-1}$). Left: dn(cc) $\Delta z$ = 0.5 m. Right: cn(cc) $\Delta z$ = 0.5 m.

















Figure 7: Simulated runoff for excess saturation in a vertical soil column using different vertical
discretizations.













Figure 8: The total number of Newton steps for excess saturation in a vertical soil column using different

vertical discretizations.




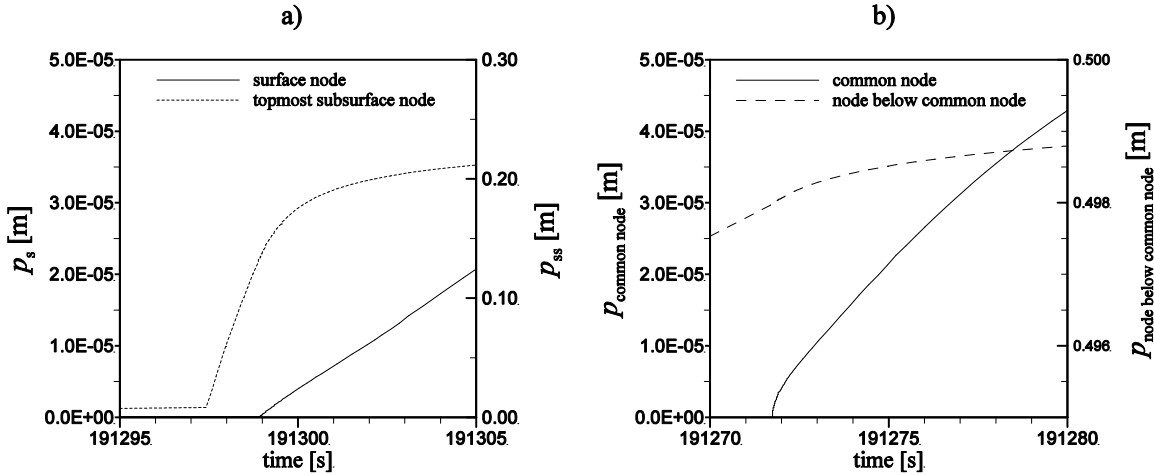

8Ɛ₀

Figure 9: Changes in pressure heads near the surface-subsurface interface for excess saturation in a vertical
soil column. Left: dn(cc) $\Delta z = 0.5$ m. Right: cn(cc) $\Delta z = 0.5$ m.











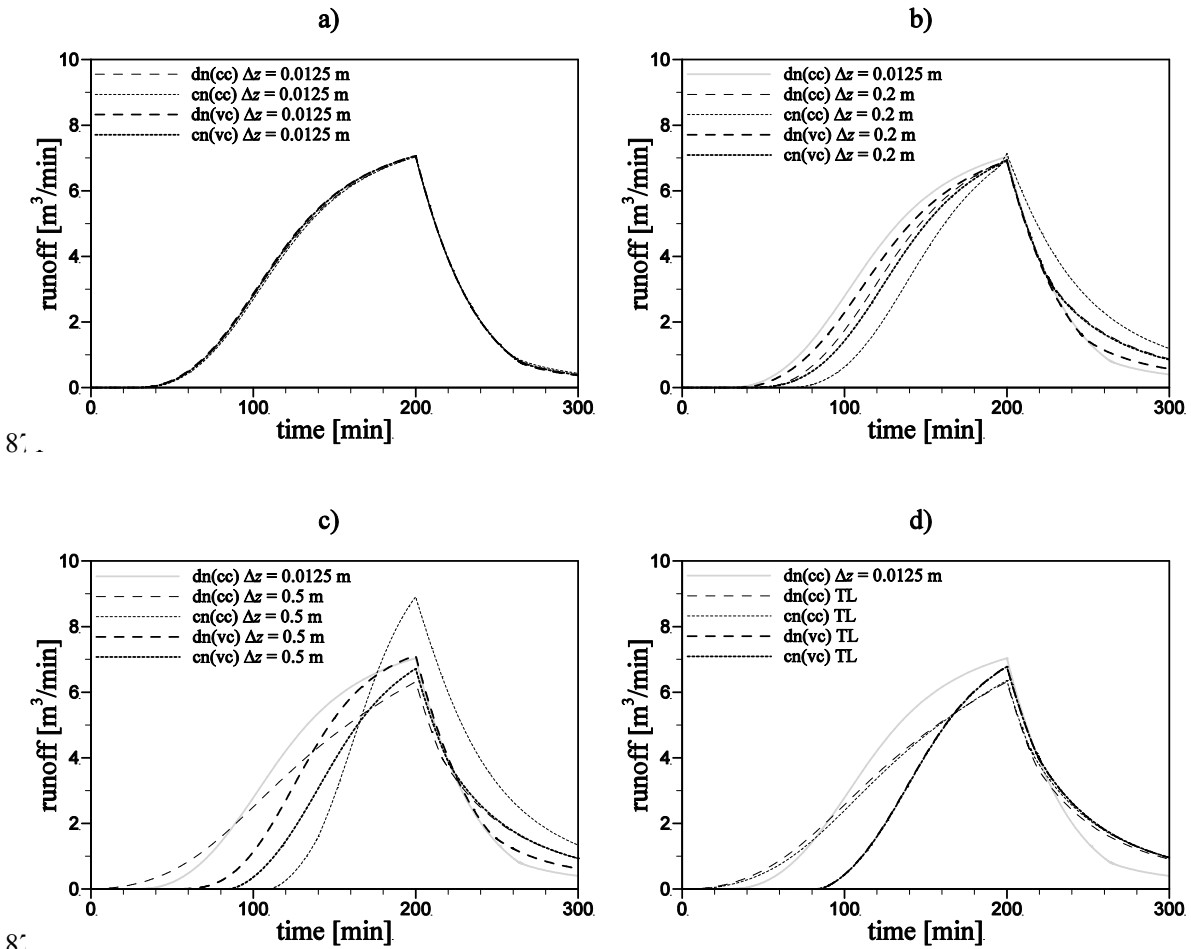

Figure 10: Outflow response for excess infiltration on a hill slope using different vertical discretizations.








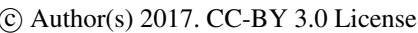



Figure 11: The total number of Newton steps for excess infiltration on a hill slope using different vertical

discretizations.

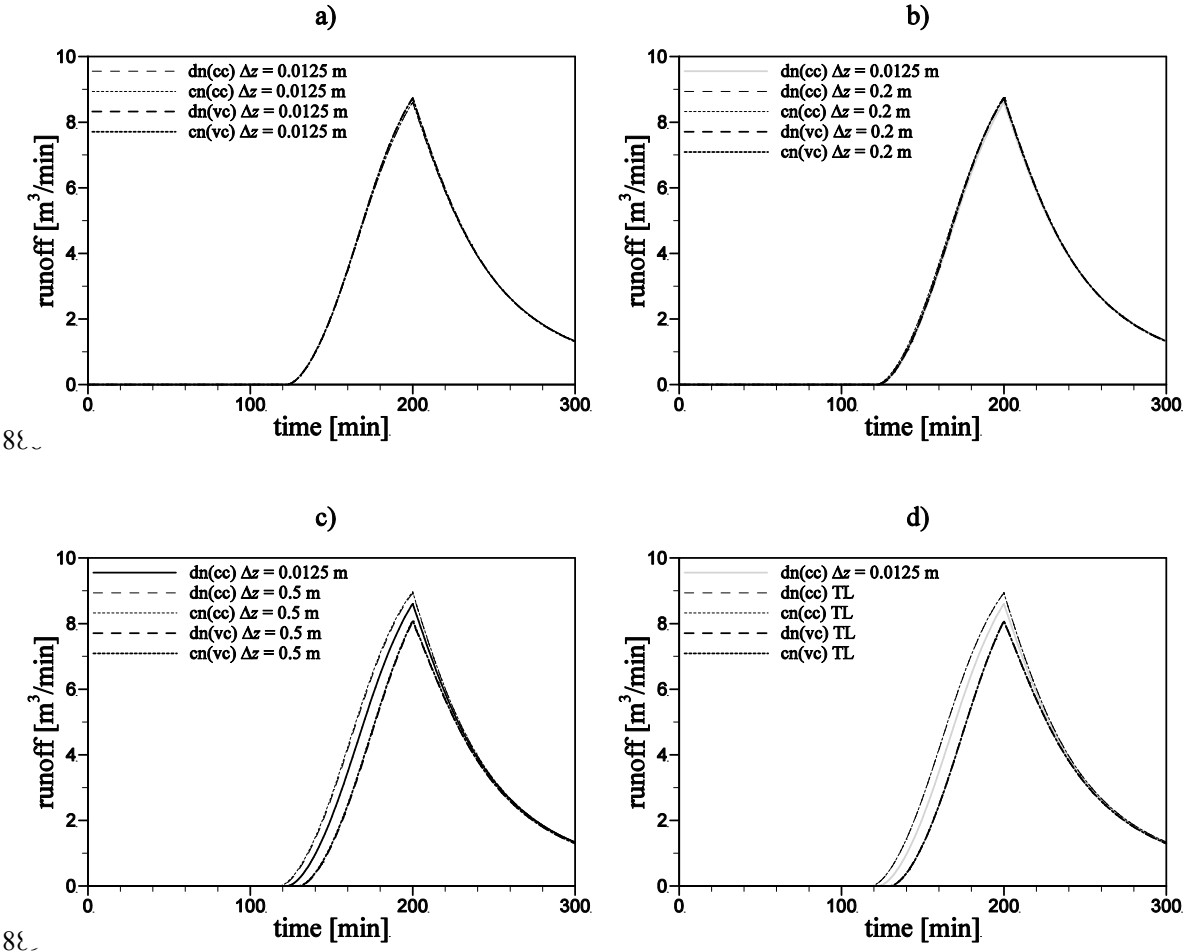

Figure 12: Outflow response for excess saturation on a hill slope using different vertical discretizations.



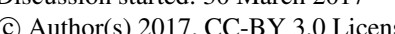



Figure 13: Number of Newton steps for excess saturation on a hill slope using different vertical discretizations.






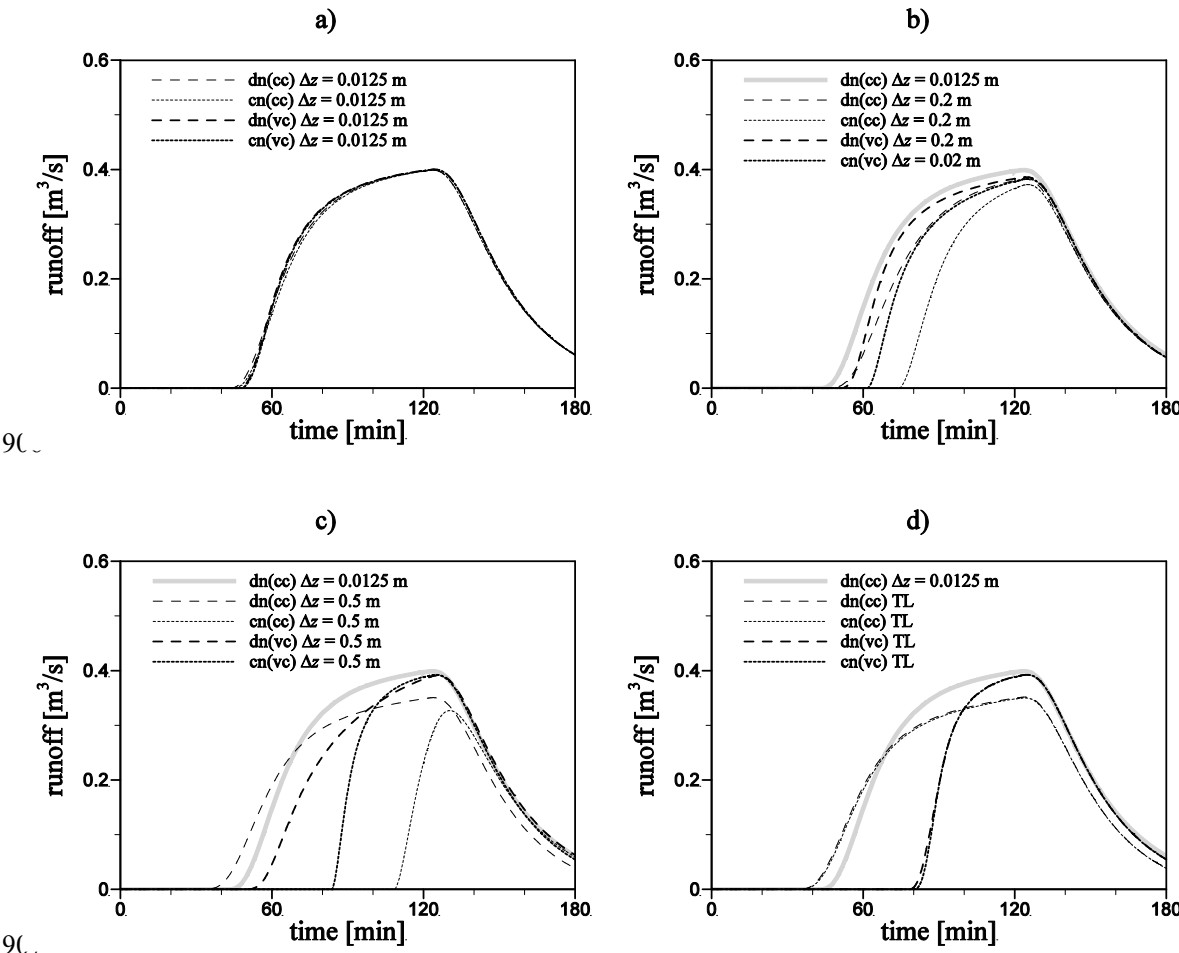

Figure 14: Outflow response for flooding an unsaturated hill slope using different vertical discretizations.













Figure 15: Number of Newton steps for flooding an unsaturated hill slope using different vertical
discretizations.








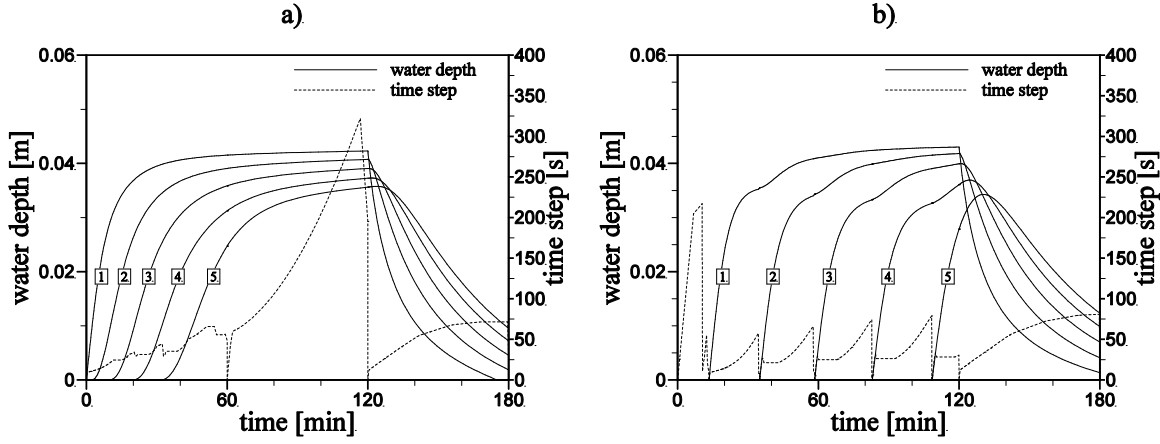

92̲

Figure 16: Response in water depth at the five surface nodes (numbered from upstream to downstream) for
flooding an unsaturated hill slope. Left: dn(cc) $\Delta z = 0.5$ m. Right: cn(cc) $\Delta z = 0.5$ m.







