# Peer review of "Hydrol. Earth Syst. Sci. Discuss., doi:10.5194/hess-2017-168, 2017 Manuscript under review for journal Hydrol. Earth Syst. Sci."

_Hydrology and Earth System Sciences, 2017_

## Referee Comment (RC1) · Anonymous Referee #1 · 12 May 2017

I have carefully read the manuscript called "A consistent implementation of the dual node approach for coupling surface-subsurface flow and its comparison to the common node approach" by Rob De Rooij. This paper raises important issues regarding the application of integrated hydrological models through the examination of the possible influence of the coupling strategy and the vertical discretization. It especially investigates the following scientific questions (i) what is the proper coupling length to be used for the so-called dual node approach; (ii) how to formulate the dual node approach to conserve the physically based nature of the model; (iii) how does the coupling strategy influence the simulated dynamics when the vertical resolution is coarsened and (iv)

how do the common node and the dual node approaches compare on synthetical test cases.

Before going to my comments of the paper, I want to stress out that these issues are critical and barely discussed in the integrated hydrologic modeling literature. Integrated hydrologic models are more and more used to investigate hydrologic behaviors but the questions of the appropriate scale, spatial resolutions (both horizontal and vertical), the crucial modeling choices that are to be made (coupling length for instance) and their effect on the simulated dynamics are too often forgotten although in my opinion of primary importance. I especially believe that there is a need to keep the physical meaning of integrated hydrological models through the use of appropriate spatial resolutions. This point is made very clear in the paper and is in a way the starting point of the research presented.

The consistent dual node approach proposed in the paper is clearly exposed and is a way to properly account for infiltration, especially in partially ponded cells. This approach for coupling allows preserving the physics of infiltration across the land surface if numerical parameters and spatial resolution are chosen adequately. A detailed analysis on the surface and subsurface pressure values, on the infiltration flux and on the time to ponding is provided. This analysis demonstrates the added-value of this method mainly (and only?) to describe the infiltration excess process. Although the issues tackled are of interest and the method proposed seems appropriate, I have serious concerns with the paper and I am not sure that the material presented is enough for a research paper. It seems that the added value of the approach proposed is not so important compared to the classical coupling approaches if the classical approaches are used in a relevant way. I hope that the following comments will somehow help improving the manuscript and maybe help in the publication process.

Major comments:

(1) One of my major concern deals with the fact that most of the conclusions of the research proposed in this paper are not novel and already documented in the literature. For instance, it has already been demonstrated that when using a proper discretization both coupling approaches gives very similar results and that a relatively small coupling length needs to be used with the dual node approach to conserve the physical meaning. It is true that integrated models tend to be used out of their proper application domain with coarse vertical discretization but it is more than intuitive that the vertical resolution should be small to properly capture the non-linear dynamics of infiltration fronts (especially when infiltration excess occurs). If the integrated models are properly applied, most of the questions that are tackled in the paper are not a problem anymore. In a way, the paper aims at determining which method is the less inaccurate (see line 554 to 556) when using a coarse vertical discretization, which is in a way irrelevant as both approaches are acceptable when using a proper resolution. These comments are illustrated through the conclusion that is short and not so much informative.

(2) The second main concern is linked to the tone and the phrasing of the paper that are not always adapted especially when reference models of the literature – i.e. Hydrogeosphere, MODHMS or Parflow – are criticized. I acknowledge that the coupling in Parflow is not well described in Kollet and Maxwell (2006) and that as a consequence some important aspects of Parflow turn out to be unclear. But I don't feel like there is a need to point out in details what the author think is not done properly by others. Once again, if an integrated model is used carefully with proper discretization and coupling length, it will produce consistent (with the physics) results regardless if it is a common node or a dual node approach. As a consequence, it is preferable to highlight what the consistent dual node approach brings than to denigrate the other approaches. I think that part 5 should be removed or at least strongly modified.

(3) I have serious concern about the result regarding the numerical efficiency. First I don't understand the arguments presented at the beginning of the part 7.2 that directly link the infiltration rate and the gradient across land surface with the numerical efficiency. It is a problem for me as all the following discussion on the efficiency is related

to that argument. I feel like this point should be explained better. Moreover, the efficiency of the resolution is highly linked to the numerical procedure (numerical scheme, time integration, …) that is used to solve the common node approach. In the paper by De Rooij (2013) it is explained that the model uses a dual node approach. But the common node approach is not described. Either I missed something or this should be detailed somewhere so that the reader can have all the needed information. Finally, for some test cases the difference in the number of Newton iteration is rather limited when using a proper discretization and coupling length making it difficult to say in a general way that the dual node approach is more efficient that the common node approach.

(4) Regarding the efficiency, I also believe that the tighter the coupling, the more difficult the resolution will be. Considering the experience I have in the domain, it is much harder to impose continuity through a common node type of approach than to impose a first order coupling through a dual node approach (if the numerical resolution is the same). As a consequence, it is for me logical that convergence is harder to obtain for some test cases with the common node approach.

(5) The paper is quite clear but some parts are too long. This makes the paper sometimes hard to read. Part 4 is an example. This part is very long and the first conclusions are deceiving – i.e the proper implementation has already been proposed by other (Line 240) and the proposition of a numerical trick to properly implement dual node in vertex-centered scheme (line 256 to 259). Maybe this can be improved.

(6) The part that presents the results is also hard to follow. I believe that there are too many test cases presented and that all of them are not needed. The saturation excess test cases may be removed as they are only illustrative for the efficiency. Maybe only the infiltration excess should be kept as it is for this process that the added-value of the method proposed is the most important. The consequence of multiple test cases per hydrological processes is that the reader has to jump from one figure to another which is not convenient at all. The number of figure presenting the results is also quite high.

(7) Regarding hydrological processes, it seems that the differences between both approaches are very small when dealing with the saturation excess process, which is the dominant process of streamflow generation in most temperate region. The main problems/conclusions are linked to the infiltration excess process. The findings for both processes are rather limited as (i) for saturation excess both approaches are OK and (ii) it is well-known that using the Richards equation infiltration excess cannot be properly capture with a 20 cm or a 50 cm resolution.

(8) The coupling between surface and subsurface strongly depends on the numerical schemes use for resolution. This point is clear on the paper (especially through the explanations related to figure 1) but the paper – although using 2 different schemes – is not exhaustive. Some published models using other resolution schemes are built using a properly implemented dual node approaches and this point should be fairly mentioned somewhere.

(9) I am a bit uneasy with the concepts of elegance and generality when considering physically-based modelling. In my opinion, the main question is whether the modelling approach chosen allows for a proper description of the physics considered. I believe that it is an endless debate to determine which approach is the more elegant or the more general and I would suggest the author to remove the sentences related to that and focus on the accuracy and/or the efficiency that are can be somehow measured.

Other comments:

- Some parts of the paper are only about interpretation and as a consequence are very subjective. See for instance from line 274 to line 283.

- Line 45: hillslopes not hill slopes

- Line 50: the reference paper for CATHY is rather Camporese et al, WRR, 2010 than Weill et al, AWR, 2011.

- Line 60: the interface is not always saturated. Its property is constant but saying that

it is always saturated can be misunderstood regarding the infiltration process.

-From line 191 to line196: this part is not clear and needs to be improved. To my knowledge and in most of the integrated models mentioned in the paper, when a cell is not ponded, all the rainfall infiltrates. When the cell is ponded or partially ponded, infiltration occurs under the ponded area. I agree that infiltration under the non-ponded fraction of a partially ponded area should be theoretically accounted for, but the sentences in the paper could lead to misunderstandings.

- Line 223: I don't understand why it is mentioned here that the surface head can be used as a Dirichlet boundary condition. I agree that it can be done but not in the context of a coupling through a dual node approach. Maybe this is linked to the implementation of the common node approach.

- Line 326: typo - Figure 1c

- Line 365-368: Repetition of things already said from line 274 to 283

- Line 395-397: I quickly checked in de Rooij et al (2013) and this paper only describe the dual node approach for coupling. Some results with the common node approach are presented later in the paper. The way the common node approach is implemented should be presented somewhere.

-Line 464 to 478: this part does not bring anything to what is already well known and described in the literature. Just say that the reference is computed using a fine resolution.

- Line 498-500: Please explain before in the paper how the inconsistent dual node approach was implemented.

- It is strange that figure 2 d and 4d shows so different results. We would expect that the behavior between different coupling approach/resolution provides same trends regarding the reference and it's not the case. Can you explain?

- Test cases with excess infiltration: even though the dual node approach displays "more desirable behavior" (line 521), the results with coarse discretizations are far from the reference. Meaning that a consistent implementation of the dual node approach is not sufficient enough if the resolution is not well chosen.

- Figure 10 c and 10 d: it is hard to say who the best is between the common node and the dual node. Needs to be discussed.

- Figure 13: why is there so much difference for this test case only? When the discharge are so close and match pretty well, the efficiency seems very different between the coupling approaches.

- Line 538-539 (excess infiltration): all the simulations are far from the reference. The argument presented in this sentence is not valid in my opinion.

- Line 553: typo "understimates or overestimates"

- Line 671: Figure 9 not 10

- Line 635: Figure 6 not 7

---

## Referee Comment (RC2) · Anonymous Referee #2 · 16 May 2017

R. deRooij (RdR) presents the dual node approach for coupling surface and groundwater flow including a comparison to the common node approach and other dual node approximations based on synthetic numerical experiments and also numerical measures (i.e. number of non-linear iterations).

I have two major points of concern with the manuscript. While I like and appreciate the effort by RdR to clarify general misperceptions and confusion of different common and dual nodes approaches, the manuscript reads more like a reckoning with numerical, hydrologic scientific software than a research paper. It is important to keep in mind

that we are dealing with a highly non-linear problem ultimately cast in discrete mathematics that a computer can understand. As such there will always be ambiguities and errors. For example, I was always wondering, how these models handle the following situation. Imagine the following thought experiment of model with a cell-centered grid, where the top layer is just under tension saturation. Adding an incremental amount of water will switch the pressure value at the cell center from some negative value to $\sim$dz/2. A dual node right at the land surface interface would switch from some negative value to $\sim$0. In both cases surface runoff is initiated. Thus, there is something like a discontinuity in pressure due to the discrete mathematics, which will lead to errors under both excess infiltration and saturation conditions for both the dual and common node approach, which can only be resolved with very high spatial discretization. This can be nicely seen, in my opinion in the results of the numerical experiments presented here and have been shown before in publications related to the simulation of coupled groundwater-surface water flow and the development of integrated hydrologic scientific software. Looking at the results presented here, these types of problems are still not resolved by the proposed dual node approach, and probably never will be because of the limitations of discrete mathematics.

Therefore, because of numerical aspects, it is also not appropriate to compare directly the non-linear iterations for both coupling schemes. The common and dual node implementation are different discrete approaches that of course will exhibit different non-linear convergence, and, second, it is not clear from the presentation how the common node approach has been implemented by RdR.

My second concern is related to the RdR's dual node approach, which is not novel. As the author acknowledges himself that "Nonetheless, their [An, H., and S. Yu (2014)] approach is actually a properly implemented dual node approach practically similar to the one proposed in this paper." Thus, it appears that main contribution of the manuscript is the discussion of the difference between the common and dual node approach and clarification of some of the applied concepts in different scientific hydrologic software.

While I feel this is a valuable contribution to the scientific literature, the manuscript requires major revisions and a more objective discussion. After all, for example, figure 2 suggests that for coarse spatial resolution both the common and dual node approach are quite far off the reference simulation. But in the past ten years or so, model implementations improved and a spatial discretization of 0.5m at the land surface is rarely used in todays models that I read about.

---

## Author Comment (AC1) · 16 May 2017

I appreciate that Anonymous Referee #1 has carefully read the manuscript and I acknowledge that certain concerns raised by the referee should be addressed if this manuscript is to be revised.

In case of a revision, I will comment in more detail in how these concerns are addressed (i.e. one by one). However, at this stage of the peer review process I would like to focus on those concerns with which I do not agree as well as those to which I would like to add my opinion.

Overall I am a bit worried that the comments of Referee #1 can be interpreted as a rejection of the manuscript. That is also the reason why I think that going through all the concerns as raised by Refereee #1 one by one is not very useful at this point.

Referee # 1 is not sure that the material is enough for a research paper. The Referee provides two reasons for this: 1) The research is not novel and its conclusions are already documented in the literature. 2) The differences between the coupling approaches are irrelevant because if properly implemented the different coupling approaches work reasonably well.

Regarding the first reason, the Referee point out that it has already been demonstrated that both coupling approaches yield similar results if a proper discretization is used. However, the fact that it is known that both approaches will yield similar results under certain conditions is acknowledged in the introduction by referencing Ebel et al. (2009) in line 69-70. Moreover, this fact is not presented as a new insight or at least this was not my intention (In a revision I can make some changes to make this more clear).

More importantly, however, it is not true in a general sense that that the conclusions of this paper are not novel. Namely:

1) Following existing literature, the common node approach is typically presented as the more general and more elegant approach, because contrary to the dual node approach, there is no need to specify a coupling length that lacks a physical meaning. In my work, I explain that the coupling length is not related to an unphysical separation between the surface and the subsurface, but to the vertical discretization of the topmost subsurface cells. Also I explain that the common node approach is only in agreement with the principle of head continuity if the topmost cells are very thin. Subsequently I present the case that if the dual node approach is properly implemented then it is actually more general as well as more elegant then the common node approach. These insights that conflict with the common consensus are novel and have not been discussed in existing literature.

[Figure]

2) I present a different conceptualization of the dual node approach in terms of a first-order approximation of the vertical hydraulic gradient at the land surface which does not rely on introducing a distinct interface between the surface and the subsurface. This is a new insight.

3) Most studies that have compared the common and dual node approach have been based on improperly implemented dual node approaches. As such the findings of my manuscript are significant.

Maybe, I misunderstand the Referee. It may be the case that the Referee is not so much claiming that my manuscript does not contain new insights, but is instead making the argument that these insights although novel are irrelevant. The Referee makes the valid point that in order to capture the infiltration fronts the vertical discretization has to be small and it is true that for sufficiently fine vertical discretizations the different coupling approaches work reasonably well. So from an application point of view the findings in my manuscript are indeed somewhat irrelevant if and only if the vertical discretization is sufficiently fine. However, there are some important reasons why my findings are still relevant:

1) When choosing a coupling approach, the existing literature may point to the common node approach as the most obvious choice. My work provides some alternatives insights into which approach is to be preferred.

2) An understanding of how these approaches work is relevant from a learning as well as a theoretical perspective.

3) There exists models in which the vertical discretization is not very fine. Even if those models violate the requisites for simulating steep infiltration fronts, it is still important to understand why different coupling approaches yield different simulation results.

4) If the dual node approach can be implemented more properly at no cost and may provide some gains in efficiency and accuracy, I fail to see why one would like to stick

with inconstant approaches just because they don't pose a problem as long as the vertical discretization is sufficiently fine.

5) Except for accuracy, there is also the question of efficiency. Even if those differences become smaller when using a finer discretization, the overall findings seems to indicate that the dual node approach is more efficient than the common node approach [This seems to correspond with the experience of the Referee].

I also want to point out that the issue of efficiency as discussed in my work is actually quite novel. In Ebel et al. (2009) efficiency is linked to the tightness of the coupling, but exactly why a tighter coupling is less efficient remains unclear. I think, that I provide additional insights into why the common node approach is less efficient (i.e. faster changes in water depth at the moment of ponding]. I agree, however, that I may need to explain this better. In any case, the issue of efficiency in surface-subsurface models is typically ignored in most of the existing literature.

Finally, I want to respond to the Referee's concerns about the tone and phrasing in the paper regarding other models. To some extent I can try to change the tone. However, to highlight the advantages of a properly implemented dual node approach, I have to say something about alternative approaches. I am not directly convinced I could skip this altogether. It is also not my intention to denigrate other approaches, but the approaches simply compare as they do and I think that these comparisons are important. For example, the inconstant dual node approaches can easily be modified. In MODHMS, one could simply turn off the adapted pressure-saturation relationship. I think that is a valuable insight.

ParFlow is an important model to reference as it was the first model to implement a common node approach. Moreover, it is a popular and powerful code. However, I cannot simply reference the existing literature and then give the configuration of the common nodes in this model. Almost anyone will point out that this configuration does not simply follow the literature (there does not exist a clear picture and explanations are

contradictory) and subsequently one may conclude that the common node approach as implemented in my work does not resemble the approach in ParFlow.

---

## Author Comment (AC2) · 16 May 2017

I appreciate that Anonymous Referee #2 feels that my research is a valuable contribution to the scientific literature.

I agree with the Referee that highly non-linear problems can only be solved using a very high spatial resolution (and in fact also requires a very fine temporal resolution). It is true that the proposed dual node approach does not resolve this requirement and I agree that the requirement of a very high resolution will probably never be solved. However, in my opinion this not mean all the errors and ambiguities associated with

numerical limitations are the same. More precisely, as explained in my manuscript the common node approach violates the principle of head continuity at the land surface if the topmost cells are not sufficiently thin and inconsistent dual node approaches also inhibit inconsistencies (for example dz/dz is not unity) which can only be removed by using very thin cells and very small coupling parameters. The proposed dual node approach is an improvement as the approach itself does not require a very fine discretization to be consistent. Of course the vertical hydraulic gradient at the land surface as computed with the proposed dual node approach is only solved sufficiently accurate when using a very fine spatial resolution. But a loss of accuracy is not the same as a loss of consistency. It is less serious and this is shown by simulation results on coarser grids.

I do not agree that it is inappropriate to compare the non-linear iterations of the coupling schemes. Rather than reporting the CPU times, I have chosen to compare the number of these iterations which in essence is the same. I think it is completely fair to subject the approaches to a comparison in efficiency as long as both approaches are subjected to the same error norms and time-stepping parameters. The only problem, I admit is to make this completely transparent without a lengthy explanation of the model code. But, this lack of transparency is partially compensated by showing the rate of changes in water depth at the moment of ponding which are different depending on the coupling scheme. I think it is important to illustrate that the number of iterations can be linked to this rate as it explains why the dual node approach can be more efficient.

Since I acknowledge that the coupling approach of An and Yu (2014) is a properly implemented dual node approach, I don't really understand the concern of the Referee considering the fact that the proposed dual node approach is not novel. To the best of my knowledge, I do not claim to have invented this approach. Instead, I point out its advantages.

In combination with the comments of Referee #1, I agree with Referee #2 that I should add some information about how the common node approach is implemented. Also I

agree that I could try to change the tone a bit. It is / was not my intention to denigrate other models. Instead I tried to be precise and to clarify the differences between them.

I would also like to discuss a bit further the thought-experiment of the Referee. So we consider the dual node approach, a topmost subsurface cell under almost fully saturated and the addition of an incremental amount of water. What happens in terms of changes in pressure head and the initiation of runoff depends on the rate at which the amount of water is supplied (i.e. the infiltration rate) and the vertical hydraulic conductivity of the subsurface (i.e. infiltration rate giving rise to either excess saturation or excess infiltration) as well as the rill storage height. As explained towards the end of section 4, under the conditions of excess infiltration ponding starts when the pressure head in the topmost cell is smaller than zero. Saturation in this cell will be reached if the water depth stays above zero for some time. Similarly, it can be shown (last part of section 4) that under the conditions of excess saturation ponding starts when the pressure head is above zero but below dz/2. In general runoff will be initiated if the water depth exceeds the rill storage height. Under no circumstances, however, will the pressure head in the topmost cell jump from negative to dz/2. What does happen is that topmost cell when p>0 can only accommodate additional water volumes by means of the specific storage such that that the rate of change in pressure head may be very fast.

In general, the initiation of ponding as well as runoff are sudden changes in the system that can be regarded as discontinuities and which are extremely difficult to solve. If ponding starts there is a sudden extra storage capacity at the land surface. If runoff starts there is a sudden introduction of additional surface flow terms. This activation / deactivation of storage and flow terms is a challenge and again I agree with the Referee that these challenges are difficult to solve and that my manuscript does not offer any solutions for these. However, that would set a high bar to cross.

---

## Author Response (AR1)

*Response to the reviewers*

*I appreciate very much the comments that were very helpful in improving the manuscript. Below I have copied all the comments and have inserted my replies in italics. I believe that this revision adequately addresses most concerns. Nonetheless, since this is a major revision, minor revisions may still be needed. I hope that the Reviewers will be available for a subsequent revision round.*

*Before addressing the comments one-by-one, I would like to address thoroughly the main concern of the reviewers. It seems that the main concern of the reviewers is that the results and conclusions as presented in the manuscript are either not novel or irrelevant. Later on I will refer to the response to this main concern as the main response.*

*Main response:*

*To address main concern of the reviewers, I have revised the manuscript in the following ways. Firstly, I have changed the title. Following the comments, I realized that the original title is misleading (A consistent implementation of the dual node approach for coupling surface-subsurface flow and its comparison to the common node approach). Namely, it may imply that this implementation is a novelty which is not the case. The new title is: New insights into the differences between the dual node approach and the common node approach for coupling surface-subsurface flow. Secondly, I have revised the manuscript thoroughly to make clearer what those insights are. Also, I have decreased the number of model scenarios and figures. I hope that this will help to keep more focus on the most important insights provided in this study and to make the manuscript easier to read.*

*Nonetheless, I disagree that that most of the conclusions in this study are not novel. To convince the Reviewers, I here summarize the novel insights provided in the revised manuscript:*

*1)*
*According to commonly held views the common node approach is more physically-based. This view is based on the idea that the dual node approach introduces an additional parameter in the form of a coupling length. Typically, this coupling length is thought of as the thickness of layer in between the surface and the subsurface domain. I show that the dual node approach can be formulated such that this coupling length is fully defined by the grid topology. This results in what I call a consistent dual node approach. In this approach, the coupling length is not a non-physical model parameter.*

*2)*
*In comparison to the common node approach the head continuity as implemented in the consistent dual node approach is more correctly formulated in the consistent dual node approach*

*3)*
*The manuscript explains in detail the comparison between the common node approach and the consistent dual node approach. Such a comparison has not been published as far as I know. Instead, most comparison studies between the dual node approach and the common node*

*approach are based on alternative dual node approaches. The comparison in this study considers accuracy as well as numerical efficiency. I think that this comparison is a valuable contribution to the scientific literature.*

*4)*
*Although the scheme is not new, the scheme is typically not being recognized as a dual node approach. Instead, it is taken as being equivalent to the common node approach (An and Yu, Kumar et al.). Even if it is recognized as a dual node scheme, it is not recognized that this is a particularly advantageous scheme (Panday and Huyakorn). This study shows that models that already use this scheme should display some advantages in terms of accuracy as well as efficiency with respect to models that use the common node approach.*

*5)*
*The manuscript explains that to understand how the approaches work, it is important to consider the meaning of nodal values. Although, this may seems trivial to a numerical modeler, it seems that this point is often overlooked. Moreover, the consistent dual node approach is derived in the manuscript from basic flow equations using finite differences. Again, this may seem trivial to a numerical modeler, but I think it is important because it leads to a conceptualization of the dual node approach, which is very different from the one in existing literature. Namely, it is not an approach that needs an additional non-physical model parameter. The manuscript also provides an explanation how the dual nodes can be separated in a vertex-centered scheme to obtain a consistent dual node scheme.*

*Except for not presenting new results or conclusions, another comment is that the differences between the approaches are irrelevant because they all yield accurate results if the spatial resolution is chosen carefully.*

*It is indeed true that all coupling approaches can yield accurate results if the vertical discretization is sufficiently fine and if the coupling length (when using a dual node approach) is carefully chosen. I made revisions such that this point is acknowledged clearer and to be more objective. It is also true that the advantages in accuracy are limited to simulating excess infiltration using a relatively coarse grid. Thus it is true as mentioned by Reviewer 1 that problems related to a difference in accuracy are not a problem if the model setup is defined carefully. However, in my opinion this does not mean that the difference in accuracy when using a relatively coarse resolution is not interesting or irrelevant. In essence, from a practical point of view, I can understand the argument why this could be irrelevant. Namely, one would wish a very fine vertical discretization irrespective of the chosen coupling approach to simulate accurately the movement of saturation fronts. But there are different coupling schemes in use today to solve surface-subsurface flow. I think it is important to understand when and why these schemes yield different results. If scientists cannot answer this question, I think that that would be a problem.*

*Moreover, the idea that a dual node approach can be more accurate than a common node approach is new and the current consensus is very different. Namely, the commonly held view is that the maximum accuracy of the dual node approach is reached when it mimics a common node approach. Also, studies based on models that use a consistent dual node approach (i.e. An*

*and Yu, Kumar et al.) did not recognize that there is a difference in accuracy with respect to the common node approach. In the study of An and Yu it was found that their model is less sensitive to the vertical discretization in comparison to ParFlow (which uses the common node approach). However, they did not recognize that this difference is related to using a different coupling approach.*

*The difference in accuracy also shows the consequence of how the head continuity is formulated in the common node approach. Namely, this formulation is only correct if the discretization is very fine and thus if a coarser vertical resolution is used, the common node approach can become less accurate than the consistent dual node approach (i.e. the formulation of head continuity in the consistent dual node approach is correct irrespective of the vertical discretization).*

*Overall, it seems that the disagreement on whether the difference in accuracy is relevant or not, depends on whether one takes a practical or theoretical view point. Obviously, I look at it from a more theoretical perspective.*

*But suppose for the sake of argument that the difference in accuracy is completely irrelevant to the scientific community. (Although as argued I do not think that is true). Then I still do not see why this would make the manuscript irrelevant in its totality. Namely, there is also the case of numerical efficiency. And the manuscript indicates quite clearly that the consistent dual node approach can be advantageous in terms of efficiency. Moreover, the case of accuracy can also be regarded as simply being an important issue to be considered when comparing two different approaches.*

Anonymous Referee #1

I have carefully read the manuscript called "A consistent implementation of the dual node approach for coupling surface-subsurface flow and its comparison to the common node approach" by Rob De Rooij. This paper raises important issues regarding the application of integrated hydrological models through the examination of the possible influence of the coupling strategy and the vertical discretization. It especially investigates the following scientific questions (i) what is the proper coupling length to be used for the so-called dual node approach; (ii) how to formulate the dual node approach to conserve the physically based nature of the model; (iii) how does the coupling strategy influence the simulated dynamics when the vertical resolution is coarsened and (iv) how do the common node and the dual node approaches compare on synthetical test cases.

*I appreciate the careful reading and interest of the reviewer.*

Before going to my comments of the paper, I want to stress out that these issues are critical and barely discussed in the integrated hydrologic modeling literature. Integrated hydrologic models are more and more used to investigate hydrologic behaviors but the questions of the appropriate scale, spatial resolutions (both horizontal and vertical), the crucial modeling choices that are to be made (coupling length for instance) and their effect on the simulated dynamics are too often forgotten although in my opinion of primary importance. I especially believe that there is a need to keep the physical meaning of integrated hydrological models through the use of appropriate spatial resolutions. This point is made very clear in the paper and is in a way the starting point of the research presented.

*I agree with the reviewer that it is important to keep the physical meaning of a model trough appropriate spatial resolutions. Indeed, this becomes a critical issue if non-linearity is significant. And this is typically the case in integrated surface-subsurface models.*

The consistent dual node approach proposed in the paper is clearly exposed and is a way to properly account for infiltration, especially in partially ponded cells. This approach for coupling allows preserving the physics of infiltration across the land surface if numerical parameters and spatial resolution are chosen adequately. A detailed analysis on the surface and subsurface pressure values, on the infiltration flux and on the time to ponding is provided. This analysis demonstrates the added-value of this method mainly (and only?) to describe the infiltration excess process. Although the issues tackled are of interest and the method proposed seems appropriate, I have serious concerns with the paper and I am not sure that the material presented is enough for a research paper. It seems that the added value of the approach proposed is not so important compared to the classical coupling approaches if the classical approaches are used in a relevant way. I hope that the following comments will somehow help improving the manuscript and maybe help in the publication process.

Major comments:

(1) One of my major concern deals with the fact that most of the conclusions of the research proposed in this paper are not novel and already documented in the literature. For instance, it has already been demonstrated that when using a proper discretization both coupling approaches gives very similar results and that a relatively small coupling length needs to be used with the dual node approach to conserve the physical meaning. It is true that integrated models tend to be used out of their proper application domain with coarse vertical discretization but it is more than intuitive that the vertical resolution should be small to properly capture the non-linear dynamics of infiltration fronts (especially when infiltration excess occurs). If the integrated models are properly applied, most of the questions that are tackled in the paper are not a problem anymore. In a way, the paper aims at determining which method is the less inaccurate (see line 554 to 556) when using a coarse vertical discretization, which is in a way irrelevant as both approaches are acceptable when using a proper resolution. These comments are illustrated through the conclusion that is short and not so much informative.

*I have addressed this concern by acknowledging in the conclusions more clearly that the advantages in terms of accuracy of the consistent dual node approach versus the common node approach are indeed limited.*

*Although I understand the reviewer's concern, I do not agree that the most conclusions are already documented nor that the differences in between the approaches are in a way irrelevant. I refer to my main response as to why I disagree.*

(2) The second main concern is linked to the tone and the phrasing of the paper that are not always adapted especially when reference models of the literature – i.e. Hydrogeosphere, MODHMS or Parflow – are criticized. I acknowledge that the coupling in Parflow is not well described in Kollet and Maxwell (2006) and that as a consequence some important aspects of Parflow turn out to be unclear. But I don't feel like there is a need to point out in details what the author think is not done properly by others. Once again, if an integrated model is used carefully with proper discretization and coupling length, it will produce consistent (with the physics) results regardless if it is a common node or a dual node approach. As a consequence, it is preferable to highlight what the consistent dual node approach brings than to denigrate the other approaches. I think that part 5 should be removed or at least strongly modified.

*I have changed the explanation of the coupling in ParFlow. However, Section 5 (now section 4.3) is in my opinion essential. This section does not aim to denigrate other models. Namely, the shortcomings of inconsistent dual node approaches have already been discussed elsewhere. As such I do not heavily criticize other models here. I merely contrast the shortcomings with the consistent dual node. Nonetheless, I have tried to change the tone and the phrasing in this section.*

(3) I have serious concern about the result regarding the numerical efficiency. First I don't understand the arguments presented at the beginning of the part 7.2 that directly link the infiltration rate and the gradient across land surface with the numerical efficiency. It is a problem for me as all the following discussion on the efficiency is related to that argument. I feel like this point should be explained better. Moreover, the efficiency of the resolution is highly linked to the numerical procedure (numerical scheme, time integration,….) that is used to solve the common node approach. In the paper by De Rooij (2013) it is explained that the model uses a dual node approach. But the common node approach is not described. Either I missed something or this should be detailed somewhere so that the reader can have all the needed information. Finally, for some test cases the difference in the number of Newton iteration is rather limited when using a proper discretization and coupling length making it difficult to say in a general way that the dual node approach is more efficient that the common node approach.

*I have made major revisions in the discussions to explain better the differences in efficiency. Moreover, I also added an explanation about how the common node approach is implemented.*

(4) Regarding the efficiency, I also believe that the tighter the coupling, the more difficult the resolution will be. Considering the experience I have in the domain, it is much harder to impose continuity through a common node type of approach than to impose a first order coupling through a dual node approach (if the numerical resolution is the same). As a consequence, it is for me logical that convergence is harder to obtain for some test cases with the common node approach.

*In the literature the difference in efficiency havs indeed been explained in terms of tight or less tight coupling (i.e. Ebel et al.).I think that my explanations of why the consistent dual node approach can be more efficient are more detailed and add some significant understanding on why the efficiency can be different. Namely, it is shown that this difference can be tied to how fast water depths are changing at the moment of ponding. These rates are different depending on the approach.*

(5) The paper is quite clear but some parts are too long. This makes the paper sometimes hard to read. Part 4 is an example. This part is very long and the first conclusions are deceiving – i.e the proper implementation has already been proposed by other (Line 240) and the proposition of a numerical trick to properly implement dual node in vertex-centered scheme (line 256 to 259). Maybe this can be improved.

*I shortened this section considerably. I have tried to re-phrase this part a bit to be clearer.*

(6) The part that presents the results is also hard to follow. I believe that there are too many test cases presented and that all of them are not needed. The saturation excess test cases may be removed as they are only illustrative for the efficiency. Maybe only the infiltration excess should be kept as it is for this process that the added-value of the method proposed is the most important. The consequence of multiple test cases per hydrological processes is that the reader has to jump from one figure to another which is not convenient at all. The number of figure presenting the results is also quite high.

*Instead of removing the saturation excess cases, I have removed the column experiments. In fact, the hillslope experiments are enough to make my points.*

(7) Regarding hydrological processes, it seems that the differences between both approaches are very small when dealing with the saturation excess process, which is the dominant process of streamflow generation in most temperate region. The main problems/conclusions are linked to the infiltration excess process. The findings for both processes are rather limited as (i) for saturation excess both approaches are OK and (ii) it is well-known that using the Richards equation infiltration excess cannot be properly capture with a 20 cm or a 50 cm resolution.

*I acknowledge more clearly that indeed, the advantage in accuracy is limited. But the fact that a dual node approach can be at least or more accurate in comparison to the common node approach is significant in my opinion as it illustrates that it matters how the head continuity is implemented. Moreover, the findings in this study contrast to commonly held views, according to which the dual node approach is only more efficient with respect to the common node approach at the expense of accuracy. Namely, the consistent dual node approach can be more efficient as well as more accurate for certain simulation scenarios. See also my main response for additional arguments why I think the difference in accuracy is actually quite relevant.*

*The point of the manuscript is not only that the consistent dual node approach can be more accurate. Efficiency is also considered. In a more general sense, the manuscript simply compares in detail the consistent dual node approach with a common node approach. The fact*

*that both approaches can yield similar results and that the differences in accuracy or efficiency are not extreme does not make this comparison study irrelevant.*

(8) The coupling between surface and subsurface strongly depends on the numerical schemes use for resolution. This point is clear on the paper (especially through the explanations related to figure 1) but the paper – although using 2 different schemes – is not exhaustive. Some published models using other resolution schemes are built using a properly implemented dual node approaches and this point should be fairly mentioned somewhere.

*I only found that the model of Kumar et al. is also in essence based on a consistent dual node approach. In addition, I also point out that CATHY as well as the model of Morita and Yen share characteristics with the consistent dual node approach.*

(9) I am a bit uneasy with the concepts of elegance and generality when considering physically-based modelling. In my opinion, the main question is whether the modelling approach chosen allows for a proper description of the physics considered. I believe that it is an endless debate to determine which approach is the more elegant or the more general and I would suggest the author to remove the sentences related to that and focus on the accuracy and/or the efficiency that are can be somehow measured.

*Corrected in the revised manuscript.*

Other comments:

- Some parts of the paper are only about interpretation and as a consequence are very subjective. See for instance from line 274 to line 283.

*Removed*

- Line 45: hillslopes not hill slopes

*Corrected*

- Line 50: the reference paper for CATHY is rather Camporese et al, WRR, 2010 than Weill et al, AWR, 2011.

*Corrected*

- Line 60: the interface is not always saturated. Its property is constant but saying that it is always saturated can be misunderstood regarding the infiltration process.

*Corrected*

-From line 191 to line196: this part is not clear and needs to be improved. To my knowledge and in most of the integrated models mentioned in the paper, when a cell is not ponded, all the rainfall infiltrates. When the cell is ponded or partially ponded, infiltration occurs under the ponded area. I agree that infiltration under the non-ponded fraction of a partially ponded area should be theoretically accounted for, but the sentences in the paper could lead to misunderstandings.

*Rephrased paragraph*

- Line 223: I don't understand why it is mentioned here that the surface head can be used as a Dirichlet boundary condition. I agree that it can be done but not in the context of a coupling through a dual node approach. Maybe this is linked to the implementation of the common node approach.

*Corrected*

- Line 326: typo - Figure 1c

*Corrected*

- Line 365-368: Repetition of things already said from line 274 to 283

*Removed*

- Line 395-397: I quickly checked in de Rooij et al (2013) and this paper only describe the dual node approach for coupling. Some results with the common node approach are presented later in the paper. The way the common node approach is implemented should be presented somewhere.

*Corrected in revised manuscript, added explanation in section 5 (numerical experiments)*

-Line 464 to 478: this part does not bring anything to what is already well known and described in the literature. Just say that the reference is computed using a fine resolution.

*This is not completely true (already known), because I compare with a consistent dual node approach which is different. But the overall idea does indeed remain the same. I have shortened the paragraph.*

- Line 498-500: Please explain before in the paper how the inconsistent dual node approach was implemented.

*I have removed this approach from the experiments (Note that it would be quite simple. Namely a simple change in elevation heads of the surface nodes).*

- It is strange that figure 2 d and 4d shows so different results. We would expect that the behavior between different coupling approach/resolution provides same trends regarding the reference and it's not the case. Can you explain?

*These figures are removed. But the difference is related to a difference in the effective rainfall rate, Namely, when simulating excess infiltration, the inconsistent dual node approach requires a water depth greater than the coupling length for top-down saturation to occur. Thus, if the effective rainfall rate is large enough then this is more likely to be the case.*

- Test cases with excess infiltration: even though the dual node approach displays "more desirable behavior" (line 521), the results with coarse discretizations are far from the reference. Meaning that a consistent implementation of the dual node approach is not sufficient enough if the resolution is not well chosen.

*I have changed the phrasing. But, the finding that the consistent dual node approach is less sensitive to the vertical discretization remains a significant insight. Indeed, I think that it can be argued well that the dual node approach displays more desirable behavior. Namely, ponding starts before the topmost subsurface node is saturated. Since this represents a value at some depth below the surface, it is logical to assume that this node should reach fully saturated conditions some time after reaching fully saturated conditions at the surface. Of course, this does not mean that the consistent dual node approach is accurate for any discretization. More in terms of comparing the two approaches I think it is fair to say that the consistent dual node approach displays more desirable behavior.*

- Figure 10 c and 10 d: it is hard to say who the best is between the common node and the dual node. Needs to be discussed.

I have removed the simulations with the coarsest discretizations. Also because Reviewer 2 stated that such a coarse discretization is rarely used.

- Figure 13: why is there so much difference for this test case only? When the discharge are so close and match pretty well, the efficiency seems very different between the coupling approaches.

*Added further and better explanation.*

- Line 538-539 (excess infiltration): all the simulations are far from the reference. The argument presented in this sentence is not valid in my opinion.

*Corrected*

- Line 553: typo "understimates or overestimates"

*Corrected*

- Line 671: Figure 9 not 10

*Corrected*

- Line 635: Figure 6 not 7

*Corrected*

Anonymous Referee #2

R. deRooij (RdR) presents the dual node approach for coupling surface and groundwater flow including a comparison to the common node approach and other dual node approximations based on synthetic numerical experiments and also numerical measures (i.e. number of non-linear iterations).

I have two major points of concern with the manuscript. While I like and appreciate the effort by RdR to clarify general misperceptions and confusion of different common and dual nodes approaches, the manuscript reads more like a reckoning with numerical, hydrologic scientific software than a research paper. It is important to keep in mind that we are dealing with a highly non-linear problem ultimately cast in discrete mathematics that a computer can understand. As such there will always be ambiguities and errors. For example, I was always wondering, how these models handle the following situation. Imagine the following thought experiment of model with a cell-centered grid,where the top layer is just under tension saturation. Adding an incremental amount of water will switch the pressure value at the cell center from some negative value to $\sim dz/2$. A dual node right at the land surface interface would switch from some negative value to $\sim 0$. In both cases surface runoff is initiated. Thus, there is something like a discontinuity in pressure due to the discrete mathematics, which will lead to errors under both excess infiltration and saturation conditions for both the dual and common node approach, which can only be resolved with very high spatial discretization. This can be nicely seen, in my opinion in the results of the numerical experiments presented here and have been shown before in publications related to the simulation of coupled groundwater-surface water flow and the development of integrated hydrologic scientific software. Looking at the results presented here, these types of problems are still not resolved by the proposed dual node approach, and probably never will be because of the limitations of discrete mathematics.

*The idea or objective of this paper is not to find a panacea for all these problems. Instead I show how the consistent dual node approach compares to the common node approach, which has not been done to the best of my knowledge. While I use similar experiments as previous studies, the results are thus novel. I use similar experiments as it is common practice to make comparisons on benchmark tests if available.*

*I have changed the tone of the paper to make it look less than a reckoning with other models. Nonetheless, to make clear that this paper contains novel insight, I do need to discuss the differences with respect to other numerical models.*

*I hope that the new figures in the manuscript will help the reviewer in finding an answer to his thought experiment. In general, the pressure head will never make an abrupt jump as long there is a specific storage greater than zero. Instead the pressure head can change very fast from 0 to a value equal to half the thickness of the topmost cell. How fast this change occurs depends on the coupling scheme as it is now explained in more detail in the manuscript. Also, when using the consistent dual node approach ponding will only start if the infiltrability is exceeded. The computation of this infiltrability depends on the vertical discretization. But in case of excess saturation, this does not matter since the ponding is merely governed by the time it takes to saturate the subsurface which depends on the initial water content and the applied flux rate.*

Therefore, because of numerical aspects, it is also not appropriate to compare directly the non-linear iterations for both coupling schemes. The common and dual node implementation are different discrete approaches that of course will exhibit different non-linear convergence, and, second, it is not clear from the presentation how the common node approach has been implemented by RdR.

*I have added an explanation of how the common node approach is implemented in the model code. However, since all the flow computations (except for the exchange flow) are identical, I think it is fair to compare the non-linear iterations. I have added a remark that the iterations do depend on how the model code is constructed. But that dependency is equal for both approaches as they are implemented in the same code. Moreover, I explain in greater detail why there are differences in efficiency. Since they can be tied to how abrupt the pressure heads are changing near the surface at the moment of ponding, I think that any model will encounter similar problems (i.e. more iterations and smaller time steps if the changes are more abrupt). Considering the concern of comparing the number of iterations, I would be interested if the Reviewer has alternative ideas of measuring the efficiency.*

My second concern is related to the RdR's dual node approach, which is not novel. As the author acknowledges himself that "Nonetheless, their [An, H., and S. Yu (2014)] approach is actually a properly implemented dual node approach practically similar to the one proposed in this paper." Thus, it appears that main contribution of the manuscript is the discussion of the difference between the common and dual node approach and clarification of some of the applied concepts in different scientific hydrologic software.

While I feel this is a valuable contribution to the scientific literature, the manuscript requires major revisions and a more objective discussion. After all, for example, figure 2 suggests that for coarse spatial resolution both the common and dual node approach are quite far off the reference simulation. But in the past ten years or so, model implementations improved and a spatial discretization of 0.5m at the land surface is rarely used in todays models that I read about.

*I have tried to strike a more objective tone and to make clearer what the paper is about. I acknowledge in the revised conclusion section that the advantage in accuracy is limited. The point that is being made in the paper is that the dual node approach should be perceived more positively in comparison to the common node approach. Namely, the common view is that a) the common node approach is more physically based, b) the common node approach is more accurate (i.e. the common view is that a dual node approach is most accurate when it mimics a common node approach), c) the dual node approach can be more efficient but at the expense of accuracy vis-à-vis the common node approach. This paper shows that this is very different when using a consistent dual node approach. Namely, in the dual node approach the head continuity is actually more properly formulated, the approach is at least at accurate as the common node approach and is often more efficient without a trade-off in accuracy. That the approach is not new (which is acknowledged in the paper) does not change the fact that these are significant new insights.*

*I also removed the spatial discretization of 0.5 m and now set the coarsest discretization to 0.2 m. Again, I do not pretend that the consistent dual node approach is always better or that it can be used with a very coarse vertical discretization. But it is interesting 
[revised manuscript text omitted]
(second scenario) on a vertical soil columnhillslope using different vertical discretizations (q_R = 10.608 md⁻¹)..

[Figure]

[Figure]

a)                                          b)

Figure 8: Simulated values at the common nodes for excess infiltration on a hillslope (second scenario) with a cell-centered scheme and $\Delta z$ = 0.0125 m.

a) Water depths. b) Pressure heads. Nodes are numbered 1-5 in the down-slope direction.

[Figure]

[Figure]

Figure 9: Simulated values for excess infiltration on a hillslope with a cell-centered scheme (second scenario) and $\Delta z = 0.2$ m. a) Water depths at the surface nodes. b) Pressure heads at the topmost subsurface nodes. Nodes are numbered 1-5 in the down-slope direction.

[Figure]

[Figure]

Figure 10: Outflow response for flooding an unsaturated hillslope using different vertical discretizations.

[Figure]

[Figure]

Figure 8: The total11: Number of Newton steps for excess saturation in a vertical soil columnflooding an unsaturated hillslope using different vertical discretizations.

[Figure]

Figure 9: Changes in pressure heads near the surface-subsurface interface for excess saturation in a vertical soil column. Left: dn(cc) Δz = 0.5 m. Right: cn(cc) Δz = 0.5 m.

[Figure]

12: Simulated values for excess infiltration (third scenario) on a hillslope with a cell-centered scheme and Δ$z$ = 0.0125 m. a) Water depths at the surface nodes. b)

Pressure heads at the topmost subsurface nodes. Nodes are numbered 1-5 in the down-slope direction).

[Figure]

124.

[Figure]

Figure 13: Simulated values for excess infiltration

[Figure]

Figure 12: (third scenario) on a

[Figure]

Figure 13: Number of Newton steps for hillslope with a

[Figure]

Figure 14: Outflow response for flooding an unsaturated hill slope using different vertical discretizations.

[Figure]

Figure 15: Number of Newton steps for flooding an unsaturated hill slope using different vertical discretizations.

[Figure]

a)                                                          b)

129~

Figure 16: Response in water depth at the five cell-centered scheme and $\Delta z$ = 0.2 m. a) Water depths at the surface nodes (numbered from upstream to downstream) for flooding an unsaturated hill slope. Left: dn(cc) $\Delta z$ = 0.5 m. Right: cn(cc) $\Delta z$ = 0.5 m. . b) Pressure heads at the topmost subsurface nodes. Nodes are numbered 1-5 in the down-slope direction).

---

## Author Response (AR2)

*Response to the reviewer*

*I appreciate the comments of the reviewer and explaining the ideas underlying the study of Kollet and Maxwell (2006). Below I have copied all the comments and have inserted my replies in italics.*

I think that the manuscript underwent improvements and got more focused, which is good. There are some structural problems (results, discussion, conclusions are in the introduction, see specific comments below) and language problems. In addition some additional clarification would help the reader.

Going back to original publication by Kollet and Maxwell (2006), I feel the paper's important point and novelty was not on a numerical approach (common or dual node or something else). The important point was the realization that the kinematic wave equation can be merged into the top flux boundary condition resulting in a free surface boundary condition at the land surface, very similar to the one used to obtain analytical solutions for a pumping test in unconfined aquifers. I feel this was really the message and as such it can be implemented in a model (numerical, perhaps even semi-analytical and analytical) in a consistent fashion to mathematically close the problem of subsurface flow as the authors wrote. Unfortunately, the numerical implementation is not very clear from the paper, but the sensitivity to the vertical discretization, which clearly is a disadvantage in case of excess infiltration, was transparently reported in the numerical test cases. Yet, this does not detract from the theoretical and practical appeal of the free surface overland flow boundary condition proposed in this paper. I feel that is something that should be pointed out in the ensuing revision of this manuscript. Note also the term "common node" approach was not introduced by Kollet and Maxwell.

*As said, I appreciate this detailed explanation about the ideas underlying the paper of Kollet and Maxwell (2006). Let me start with the terms dual and common nodes. It is true that these terms do not appear in the paper. However, they have been used to distinguish between the two coupling approaches, most notably by the people who developed the code HydroGeoSphere. In my opinion the terms are very appropriate as they are short and self-explanatory. It is true, that originally these terms apply to vertex-centered schemes. But once the cell center in a cell-centered scheme is defined as a 'node', then the terms can also be applied to cell-centered schemes.*

*In my opinion, I clearly mentioned that the key advantages (theoretical and practical) of the idea of Kollet and Maxwell (2006). Namely in the introduction, I mention that the idea avoids the introduction of a coupling length / exchange parameter. Avoiding this model parameter is an advantage as it is often believed to be a non-physical parameter. A key point of my study is, however, is that this model parameter should be a function of the vertical discretization. Moreover, as I discussed, the way of enforcing head continuity at the surface in the common node approach is somewhat problematic.*

I am surprised by the (sometimes) large difference in non-linear iterations, because in principle the
common node and dual nodes approach are very similar (which is also the reason why they can be
easily switched in the code as described in the manuscript). Which additional nonlinearity is
introduced in case of the common node approach? A more satisfactory attempt of explaining this
phenomenon would be needed in my opinion.

*In my opinion this is actually well explained in the manuscript (section 6). But I agree, that if one*
*is focusses on the similarities between the coupling approaches, the differences are somewhat*
*surprising. In fact it is not related to differences in non-linearities, but to rates at which the flow*
*variables are changing around the moment that ponding is initiated. At this time the system*
*displays a discontinuity in flow behavior as surface flow terms are about to be activated*
*(regardless of the coupling approach). The higher the rate at which the flow variables are*
*changing, the more difficult the solution will be. As discussed in section 6 and illustrated by*
*figures, the common node approach is characterized by higher rates and is thus more difficult to*
*solve. These differences in the rate only occur during a short moment but exactly at the moment*
*that the system is difficult to solve. Going a bit further, the differences in the rate of change are*
*related to the condition for ponding to start. The crux here is that the common node approach*
*requires fully saturated conditions for ponding to occur. That means that just before ponding the*
*rate of change in the pressure head is quite high.*

Provided the general comments above and more specific comments below, I am recommending
minor revisions.

68: Is Kollet and Maxwell (2006) and appropriate reference for MODHMS?

*Corrected*

100 - 127: These are results and discussion/summary/conclusions and do not belong in the
introduction section.

*In my opinion this is not the case. Namely, an important part of my study focusses on how the dual*
*node approach should be implemented and this should be clear from the introduction.*
*Furthermore that part of the work builds on the work of others (An and Yu, Panday and Huyakorn).*
*Those previous studies should thus be referenced in the introduction.*

137: ...the mesh...

*Corrected*

220: What is fp in the equation?

Line 187: the fraction of the interface that is ponded. But I noticed that the p index was in italics.
That was incorrect.

232: ...dua nodes. The infiltration…

*Rephrased*

292: I am confused and not sure what the authors wants get at. When qr = I then qr = Kz, thus pss
= 0, because there is a unit gradient at the right at the land surface. If pss < 0 then qr < I, contrary
to equation 8 because ps is not zero. In case of qr > I, again qs is not equal zero and pss < 0 is
wrong in this case. The physics and transient pressure behavior near the surface right at the onset
of ponding is more complicated and has been discussed extensively in the literature (Kutilek and
Nielsen). Storativity concepts need to be introduced if I remember correctly. If the author wants to
look at these processes in the some limit, then this should be done in a mathematically rigorous
fashion in my opinion.

*I have changed this paragraph. I think it is now more clear what I want to get at. In particular, I*
*make it more clear where the ps = 0 is coming from.*

376:

380: necessarily

*Corrected*

441: Kollet and Maxwell also showed that in their original publication, no?

*Corrected*

In figure 4 and 5, it is very difficult to distinguish between the dn and cn results.

*Corrected (using dashed lines for common node plots)*

Make sure to carefully check language and grammar, especially for typos.

[revised manuscript text omitted]